# Escape from neutralizing antibodies by SARS-CoV-2 spike protein variants

Yiska Weisblum[1†], Fabian Schmidt[1†], Fengwen Zhang[1], Justin DaSilva[1], Daniel Poston[1], Julio CC Lorenzi[2], Frauke Muecksch[1], Magdalena Rutkowska[1], Hans-Heinrich Hoffmann[3], Eleftherios Michailidis[3], Christian Gaebler[2], Marianna Agudelo[2], Alice Cho[2], Zijun Wang[2], Anna Gazumyan[2], Melissa Cipolla[2], Larry Luchsinger[4], Christopher D Hillyer[4], Marina Caskey[2], Davide F Robbiani[2,5], Charles M Rice[3], Michel C Nussenzweig[2,6], Theodora Hatziioannou[1*], Paul D Bieniasz[1,6*]

[1]Laboratory of Retrovirology, The Rockefeller University, New York, United States; [2]Laboratory of Molecular Immunology The Rockefeller University, New York, United States; [3]Laboratory of Virology and Infectious Disease The Rockefeller University, New York, United States; [4]Lindsley F. Kimball Research Institute, New York Blood Center, New York, United States; [5]Institute for Research in Biomedicine, Università della Svizzera italiana, Bellinzona, Switzerland; [6]Howard Hughes Medical Institute, The Rockefeller University, New York, United States

*For correspondence:
thatziio@rockefeller.edu (TH);
pbieniasz@rockefeller.edu (PDB)

[†]These authors contributed equally to this work

**Abstract** Neutralizing antibodies elicited by prior infection or vaccination are likely to be key for future protection of individuals and populations against SARS-CoV-2. Moreover, passively administered antibodies are among the most promising therapeutic and prophylactic anti-SARS-CoV-2 agents. However, the degree to which SARS-CoV-2 will adapt to evade neutralizing antibodies is unclear. Using a recombinant chimeric VSV/SARS-CoV-2 reporter virus, we show that functional SARS-CoV-2 S protein variants with mutations in the receptor-binding domain (RBD) and N-terminal domain that confer resistance to monoclonal antibodies or convalescent plasma can be readily selected. Notably, SARS-CoV-2 S variants that resist commonly elicited neutralizing antibodies are now present at low frequencies in circulating SARS-CoV-2 populations. Finally, the emergence of antibody-resistant SARS-CoV-2 variants that might limit the therapeutic usefulness of monoclonal antibodies can be mitigated by the use of antibody combinations that target distinct neutralizing epitopes.

## Introduction

Neutralizing antibodies are a key component of adaptive immunity against many viruses that can be elicited by natural infection or vaccination (*Plotkin, 2010*). Antibodies can also be administered as recombinantly produced proteins or as convalescent plasma to confer a state of passive immunity in prophylactic or therapeutic settings. These paradigms are of particular importance given the emergence of SARS-CoV-2, and the devastating COVID19 pandemic that has ensued. Indeed, interventions to interrupt SARS-CoV-2 replication and spread are urgently sought, and passively administered antibodies are currently among the most promising therapeutic and prophylactic antiviral agents. Moreover, an understanding of the neutralizing antibody response to SARS-CoV-2 is critical for the elicitation of effective and durable immunity by vaccination (*Kellam and Barclay, 2020*).

Recent studies have shown that related, potently neutralizing monoclonal antibodies that recognize the SARS-CoV-2 receptor-binding domain (RBD) are often elicited in SARS-CoV-2 infection

**eLife digest** The new coronavirus, SARS-CoV-2, which causes the disease COVID-19, has had a serious worldwide impact on human health. The virus was virtually unknown at the beginning of 2020. Since then, intense research efforts have resulted in sequencing the coronavirus genome, identifying the structures of its proteins, and creating a wide range of tools to search for effective vaccines and therapies. Antibodies, which are immune molecules produced by the body that target specific segments of viral proteins can neutralize virus particles and trigger the immune system to kill cells infected with the virus. Several technologies are currently under development to exploit antibodies, including vaccines, blood plasma from patients who were previously infected, manufactured antibodies and more.

The spike proteins on the surface of SARS-CoV-2 are considered to be prime antibody targets as they are accessible and have an essential role in allowing the virus to attach to and infect host cells. Antibodies bind to spike proteins and some can block the virus' ability to infect new cells. But some viruses, such as HIV and influenza, are able to mutate their equivalent of the spike protein to evade antibodies. It is unknown whether SARS-CoV-2 is able to efficiently evolve to evade antibodies in the same way.

Weisblum, Schmidt et al. addressed this question using an artificial system that mimics natural infection in human populations. Human cells grown in the laboratory were infected with a hybrid virus created by modifying an innocuous animal virus to contain the SARS-CoV-2 spike protein, and treated with either manufactured antibodies or antibodies present in the blood of recovered COVID-19 patients. In this situation, only viruses that had mutated in a way that allowed them to escape the antibodies were able to survive. Several of the virus mutants that emerged had evolved spike proteins in which the segments targeted by the antibodies had changed, allowing these mutant viruses to remain undetected. An analysis of more than 50,000 real-life SARS-CoV-2 genomes isolated from patient samples further showed that most of these virus mutations were already circulating, albeit at very low levels in the infected human populations.

These results show that SARS-CoV-2 can mutate its spike proteins to evade antibodies, and that these mutations are already present in some virus mutants circulating in the human population. This suggests that any vaccines that are deployed on a large scale should be designed to activate the strongest possible immune response against more than one target region on the spike protein. Additionally, antibody-based therapies that use two antibodies in combination should prevent the rise of viruses that are resistant to the antibodies and maintain the long-term effectiveness of vaccines and therapies.

(*Robbiani et al., 2020*; *Brouwer et al., 2020*; *Cao et al., 2020*; *Chen et al., 2020*; *Chi et al., 2020*; *Rogers et al., 2020*; *Shi et al., 2020*; *Wu et al., 2020a*; *Wec et al., 2020*; *Kreer et al., 2020*; *Hansen et al., 2020*; *Ju et al., 2020*; *Seydoux et al., 2020*; *Liu et al., 2020*; *Zost et al., 2020*). These antibodies have great potential to be clinically impactful in the treatment and prevention of SARS-CoV-2 infection. The low levels of somatic hypermutation and repetitive manner in which similar antibodies (e.g. those based on IGHV3-53 *Robbiani et al., 2020*; *Barnes et al., 2020*; *Yuan et al., 2020*) have been isolated from COVID19 convalescents suggests that potently neutralizing responses should be readily elicited. Paradoxically, a significant fraction of COVID19 convalescents, including some from whom potent neutralizing antibodies have been cloned, exhibit low levels of plasma neutralizing activity (*Robbiani et al., 2020*; *Wu et al., 2020b*; *Luchsinger et al., 2020*). Together, these findings suggest that natural SARS-CoV-2 infection may often fail to induce sufficient B-cell expansion and maturation to generate high-titer neutralizing antibodies.

The degree to, and pace at which SARS-CoV-2 might evolve to escape neutralizing antibodies is unclear. The aforementioned considerations raise the possibility that SARS-CoV-2 evolution might be influenced by frequent encounters with sub-optimal concentrations of potently neutralizing antibodies during natural infection. Moreover, the widespread use of convalescent plasma containing unknown, and often suboptimal, levels of neutralizing antibodies might also increase the acquisition of neutralizing antibody resistance by circulating SARS-CoV-2 populations (*Bloch et al., 2020*; *Al-Riyami et al., 2020*). Reinfection of previously infected individuals who have incomplete or

waning serological immunity might similarly drive emergence of antibody escape variants. As human neutralizing antibodies are discovered and move into clinical development as therapeutics and prophylactics, and immunogens based on prototype SARS-CoV-2 spike protein sequences are deployed as vaccines, it is important to anticipate patterns of antibody resistance that might arise. Here, we describe a recombinant chimeric virus approach that can rapidly generate and evaluate SARS-CoV-2 S mutants that escape antibody neutralization. We show that mutations conferring resistance to convalescent plasma or RBD-specific monoclonal antibodies can be readily generated in vitro. Notably, these antibody resistance mutations are present at low frequency in natural populations. Importantly, the use of candidate monoclonal antibody combinations that target distinct epitopes on the RBD (and therefore have non-overlapping resistance mutations) can suppress the emergence of antibody resistance.

## Results

### Selection of SARS-CoV-2 S variants using a replication-competent VSV/ SARS-CoV-2 chimeric virus

To select SARS-CoV-2 S variants that escape neutralization by antibodies, we used a recently described replication-competent chimeric virus based on vesicular stomatitis virus that encodes the SARS-CoV-2 spike (S) protein and green fluorescent protein (rVSV/SARS-CoV-2/GFP) (*Schmidt et al., 2020*). Notably, rVSV/SARS-CoV-2/GFP replicates rapidly and to high-titers ($10^7$ to $10^8$ PFU/ml within 48 hr), mimics the SARS-CoV-2 requirement for ACE-2 as a receptor, and is neutralized by COVID19 convalescent plasma and SARS-CoV-2 S-specific human monoclonal antibodies (*Schmidt et al., 2020*). The replication of rVSV/SARS-CoV-2/GFP can be readily monitored and measured by GFP fluorescence and the absence of proof-reading activity in the viral polymerase (VSV-L) results in the generation of virus stocks with greater diversity than authentic SARS-CoV-2, for an equivalent viral population size. These features facilitate experiments to investigate the ability S protein variants to escape antibody neutralization.

We used two adapted, high-titer variants of rVSV/SARS-CoV-2/GFP, (namely rVSV/SARS-CoV-2/GFP$_{1D7}$ and rVSV/SARS-CoV-2/GFP$_{2E1}$) (*Schmidt et al., 2020*) in attempts to derive antibody-resistant mutants. Virus populations containing $1 \times 10^6$ infectious particles were generated following three passages to generated sequence diversity. On the third passage, cells were infected at an MOI of ~ 0.5 and progeny harvested after as short a time as possible so as to minimize phenotypic mixing in the viral population and to maximize the concordance between the genome sequence and the S protein sequence represented in a given virion particle. Because the mutation rate of VSV is ~ $10^{-4}$ to $10^{-5}$/base per replication cycle (*Steinhauer and Holland, 1986*; *Steinhauer et al., 1989*; *Combe and Sanjuán, 2014*), we estimated that this procedure should generate a large fraction of the possible replication-competent mutants within a population size of $1 \times 10^6$. The viral populations were then incubated with antibodies to neutralize susceptible variants (*Figure 1A*). For monoclonal antibodies, viral populations were incubated with antibodies at 5 µg/ml or 10 µg/ml, (~1000 to 10,000 x IC$_{50}$) so as to minimize the number of infection events by antibody sensitive variants, and enable rapid selection of the most resistant rVSV/SARS-CoV-2/GFP variants from the starting population. For plasma samples, the possibility existed that multiple different antibody specificities could be present, that might interfere with the outgrowth of rVSV/SARS-CoV-2/GFP variants that were resistant to the most prevalent or potent antibodies in the plasma. Therefore, in these selection experiments, viruses were incubated with a range of plasma dilutions (see materials and methods). Neutralized viral populations were then applied to 293T/ACE2(B) cells (*Schmidt et al., 2020*), which support robust rVSV/SARS-CoV-2/GFP replication, and incubated for 48 hr. We used three potent human monoclonal antibodies C121, C135, and C144 (*Robbiani et al., 2020*), that are candidates for clinical development (*Table 1*). In addition, we used four convalescent plasma samples, three of which were from the same donors from which C121, C135, and C144, were obtained (*Robbiani et al., 2020*; *Table 1*). Two of these plasmas (COV-47 and COV-72) were potently neutralizing while the third (COV-107) had low neutralizing activity. A fourth convalescent plasma sample (COV-NY) was potently neutralizing but did not have a corresponding monoclonal antibody (*Table 1*).

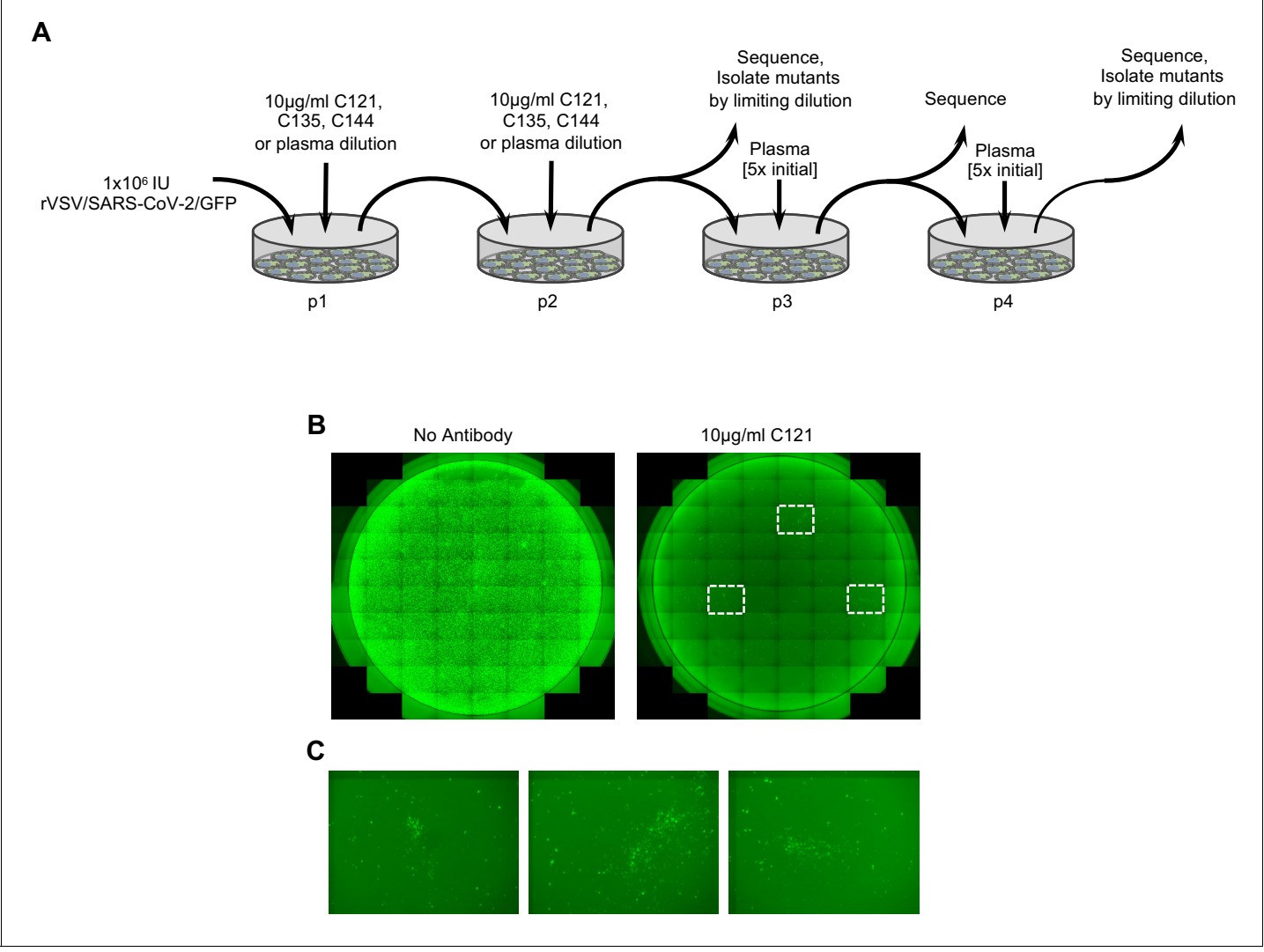

**Figure 1.** Selection of SARS-CoV-2 S mutations that confer antibody resistance. (A) Outline of serial passage experiments with replication-competent VSV derivatives encoding the SARS-CoV-2 S envelope glycoprotein and a GFP reporter (rVSV/SARS-CoV-2/GFP) in 293T/ACE2(B) cells in the presence of neutralizing antibodies or plasma. Each passage experiment was performed twice (once each with rVSV/SARS-CoV-2/GFP$_{1D7}$ and rVSV/SARS-CoV-2/GFP$_{2E1}$.) (B) Representative images of 293T/ACE2(B) cells infected with $1 \times 10^6$ PFU of rVSV/SARS-CoV-2/GFP in the presence or absence of 10 μg/ml of the monoclonal antibody C121. (C) Expanded view of the boxed areas showing individual plaques of putatively antibody-resistant viruses.

Infection with rVSV/SARS-CoV-2/GFP in the presence of the monoclonal antibodies C121 or C144 reduced the number of infectious units from $10^6$ to a few hundred, as estimated by the frequency of GFP-positive cells (*Figure 1B*) a reduction of > 1000 fold. C135 reduced infection by ~ 100 fold.

**Table 1.** Plasma and monoclonal antibodies used in this study.

| Donor | Plasma NT$_{50}$ (rVSV-SARSCoV2/GFP) | Plasma NT$_{50}$ HIV/ CCNGnLuc | Monoclonal antibody |
|---|---|---|---|
| COV-47 | 6622 | 8016 | C144 |
| COV-72 | 6274 | 7982 | C135 |
| COV-107 | 122 | 334 | C121 |
| COV-NY | 12614 | 7505 | ND |

Imaging of wells infected with rVSV/SARS-CoV-2/GFP in the presence of C121 or C144 revealed a small number of foci (~10 to 20/well), that suggested viral spread following initial infection (*Figure 1B*). In the case of C135, a greater number of GFP-positive cells were detected, obscuring the visualization of focal viral spread following initial infection. Aliquots of supernatants from these passage-1 (p1) cultures were collected 48 hr after infection, diluted in the same concentrations of monoclonal antibodies that were initially employed, and used to infect fresh (p2) cultures (*Figure 1A*). For p2 cultures, almost all cells became infected within 48 hr, suggesting the possible outgrowth of monoclonal antibody escape variants that were present in the original viral populations.

For selection in the presence of plasma, p1 supernatants were harvested at 48 hr after infection in the presence of the highest concentrations of plasma that permitted infection of reasonable numbers (approximately 10%) of cells. Then, p2 cultures were established using p1 supernatants, diluted in the same concentrations of plasma used in p1. This approach led to clear 'escape' for the COV-NY plasma with prolific viral growth in p2 as evidenced by a large increase in the number of GFP-positive cells. For COV-47, COV-72, and CO107, plasma clearly retained at least some inhibitory activity in p2. Thereafter, p3 cultures and p4 cultures were established for COV-47, COV-72, and COV-107 plasmas at 5-fold higher concentrations of plasma than were used in p1 and p2 cultures (*Figure 1A*).

RNA was extracted from p2 supernatants (monoclonal antibodies and COV-NY plasma) as well as later passages for the COV-47, COV-72, and COV-107 plasma selections. Sequences encoding either the RBD or the complete S protein were amplified using PCR and analyzed by Sanger and/or Illumina sequencing. For all three monoclonal antibodies and two of the four plasmas, sequence analyses revealed clear evidence for selection, with similar or identical mutants emerging in the presence of monoclonal antibodies or plasma in both rVSV/SARS-CoV-2/GFP$_{1D7}$ and rVSV/SARS-CoV-2/GFP$_{2E1}$ cultures (*Figure 2A–D*, *Figure 2—figure supplement 1A,B*, *Figure 2—figure supplement 2A,B*, *Table 2*). In the case of C121, mutations E484K and Q493K/R within the RBD were present at high frequencies in both p2 selected populations, with mutation at a proximal position (F490L) present in one p2 population (*Figure 2A*, *Table 2*). Viruses passaged in the presence of monoclonal antibody C144 also had mutations at positions E484 and Q493, but not at F490 (*Figure 2C*, *Table 2*). In contrast, virus populations passaged in the presence of monoclonal antibody C135 lacked mutations at E484 or Q493, and instead had mutations R346K/S/L and N440K at high frequency (*Figure 2C*, *Table 2*).

Mutations at specific positions were enriched in viruses passaged in the presence of convalescent plasma, in two out of four cases (*Figure 2—figure supplement 1A,B*, *Figure 2—figure supplement 2A,B*, *Table 2*). Specifically, virus populations passaged in the presence of COV-NY plasma had mutations within RBD encoding sequence (K444R/N/Q and V445E) that were abundant at p2 (*Figure 2—figure supplement 2A,B*, *Table 2*). Conversely, mutations outside the RBD, specifically at N148S, K150R/E/T/Q and S151P in the N-terminal domain (NTD) were present at modest frequency in COV-47 p2 cultures and became more abundant at p3 and p4 (*Figure 2—figure supplement 1A, B*, *Table 2*). Replication in the presence of COV72 or COV107 plasma did not lead to the clear emergence of escape mutations, suggesting that the neutralization by these plasmas was not due to one dominant antibody specificity. In the case of COV107, the failure of escape mutants to emerge may simply be due to the lack of potency of that plasma (*Table 1*). However, in the case of COV-72, combinations of antibodies may be responsible for the potent neutralizing properties of the plasma in that case.

## Isolation and characterization of rVSV/SARS-CoV-2/GFP antibody escape mutants

Based on the aforementioned analyses, supernatants from C121, C144, and C135 and COV-NY plasma p2 cultures, or COV47 p4 cultures, contained mixtures of putative rVSV/SARS-CoV-2/GFP neutralization escape mutants. To isolate individual mutants, the supernatants were serially diluted and individual viral foci isolated by limiting dilution in 96-well plates. Numerous individual rVSV/SARS-CoV-2/GFP$_{1D7}$ and rVSV/SARS-CoV-2/GFP$_{2E1}$ derivatives were harvested from wells containing a single virus plaque, expanded on 293T/ACE2(B) cells, then RNA was extracted and subjected to Sanger-sequencing (*Figure 3—figure supplement 1*). This process verified the purity of the individual rVSV/SARS-CoV-2/GFP variants and yielded a number of viral mutants for further analysis

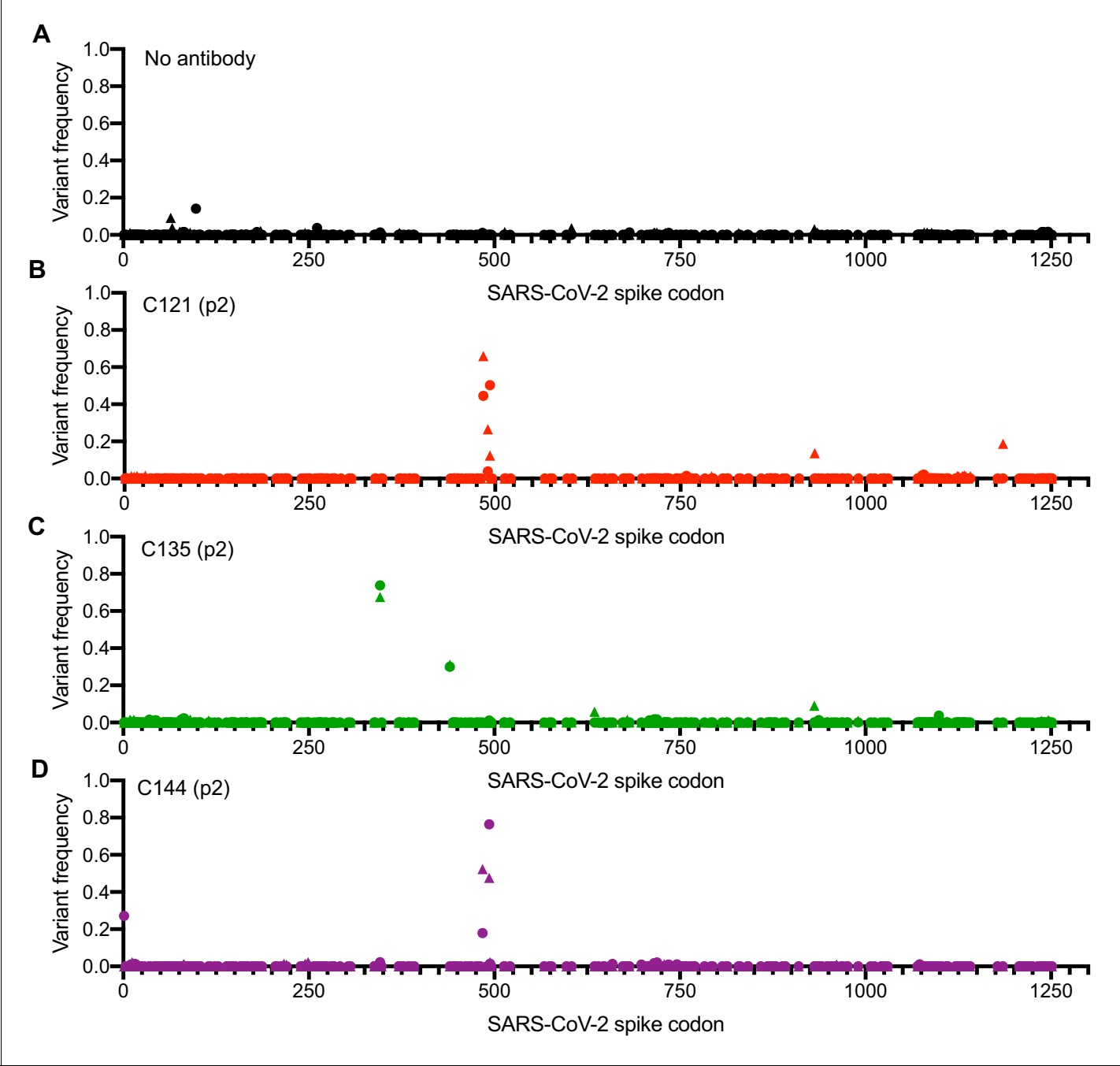

**Figure 2.** Analysis of S-encoding sequences following rVSV/SARS-CoV-2/GFP replication in the presence of neutralizing monoclonal antibodies. (A–D) Graphs depict the S codon position (X-axis) and the frequency of non-synonymous substitutions (Y-axis) following the second passage (p2) of rVSV/ SARS-CoV-2/GFP on 293T/ACE2(B) cells in the absence of antibody or plasma (A), or in the presence of 10 µg/ml C121 (B), C135 (C) or C144 (D). Results are shown for both rVSV/SARS-CoV-2/GFP variants (One replicate each for rVSV/SARS-CoV-2/GFP$_{1D7}$ and rVSV/SARS-CoV-2/GFP$_{2E1}$ - the frequency of 1D7 mutations is plotted as circles and 2E1 mutations as triangles).

The online version of this article includes the following figure supplement(s) for figure 2:

**Figure supplement 1.** Analysis of S-encoding sequences following rVSV/SARS-CoV-2/GFP replication in the presence of convalescent plasma COV-47 and COV-72.

**Figure supplement 2.** Analysis of S-encoding sequences following rVSV/SARS-CoV-2/GFP replication in the presence of convalescent plasma COV-107 and COV-NY.

**Table 2.** Mutations occurring at high frequency during rVSV/SARS-CoV-2 passage in the presence of neutralizing antibodies or plasma.

| | | Mutant frequency | | |
|---|---|---|---|---|
| | Mutation | p2 | p3 | p4 |
| Monoclonal antibodies | | | | |
| C121 | E484K* | 0.50, 0.39 | –† | – |
| | F490L | 0.23 | | |
| | Q493K | 0.12, 0.45 | | |
| C135 | N440K | 0.31, 0.30 | – | – |
| | R346S | 0.30, 0.17 | | |
| | R346K | 0.22, 0.53 | | |
| | R346M | 0.16 | | |
| C144 | E484K | 0.44, 0.18 | – | – |
| | Q493K | 0.31, 0.39 | | |
| | Q493R | 0.17, 0.37 | | |
| Plasmas | | | | |
| COV47 | N148S | 0.16, 0.14 | 0.29, 0.30 | 0.72, 0.14 |
| | K150R | 0.10 | | 0.18 |
| | K150E | 0.04 | 0.16 | 0.4 |
| | K150T | | | 0.22 |
| | K150Q | | 0.16 | 0.22 |
| | S151P | 0.1 | 0.18 | 0.2 |
| COV-NY | K444R | 0.20,0.19 | – | – |
| | K444N | 0.14 | | |
| | K444Q | 0.33 | | |
| | V445E | 0.18 | | |

*Values represent the decimal frequency with which each mutation occurs ass assessed by NGS, two values indicate occurrences in both rVSV/SARS-CoV-2/GFP$_{1D7}$ and rVSV/SARS-CoV-2/GFP$_{2E1}$ cultures, single values indicate occurrence in only one culture.

† –, not done.

(*Figure 3—figure supplement 1*). These plaque-purified viral mutants all encoded single amino-acid substitutions in S-coding sequences that corresponded to variants found at varying frequencies (determined by Illumina sequencing) in the antibody-selected viral populations. Notably, each of the isolated rVSV/SARS-CoV-2/GFP mutants replicated with similar kinetics to the parental rVSV/SARS-CoV-2/GFP$_{1D7}$ and rVSV/SARS-CoV-2/GFP$_{2E1}$ viruses (*Figure 3A*), suggesting that the mutations that emerged during replication in the presence of monoclonal antibodies or plasma did not confer a substantial loss of fitness, at least in the context of rVSV/SARS-CoV-2/GFP. Moreover, for mutants in RBD sequences that arose in the C121, C135, C144, and COV-NY cultures, each of the viral mutants retained approximately equivalent sensitivity to neutralization by an ACE2-Fc fusion protein, suggesting little or no change in interaction with ACE2 (*Figure 3B*).

We next determined the sensitivity of the isolated RBD mutants to neutralization by the three monoclonal antibodies. The E484K and Q493R mutants that emerged during replication in the presence of C121 or C144, both caused apparently complete, or near complete, resistance to both antibodies (IC$_{50}$ > 10 µg/ml, *Figure 4A,B*). However, both of these mutants retained full sensitivity (IC$_{50}$ < 10 ng/ml) to C135. Conversely, the R346S and N440K mutants that emerged during replication in the presence of C135 were resistant to C135, but retained full sensitivity to both C121 and C144 (*Figure 4A,B*). The K444N, K444T, V445G, V445E, and V445L mutants that arose during replication in the presence of COV-NY plasma conferred partial resistance to C135, with IC$_{50}$ values ranging from 25 to 700 ng/ml, but these mutants retained full sensitivity to both C121 and C144

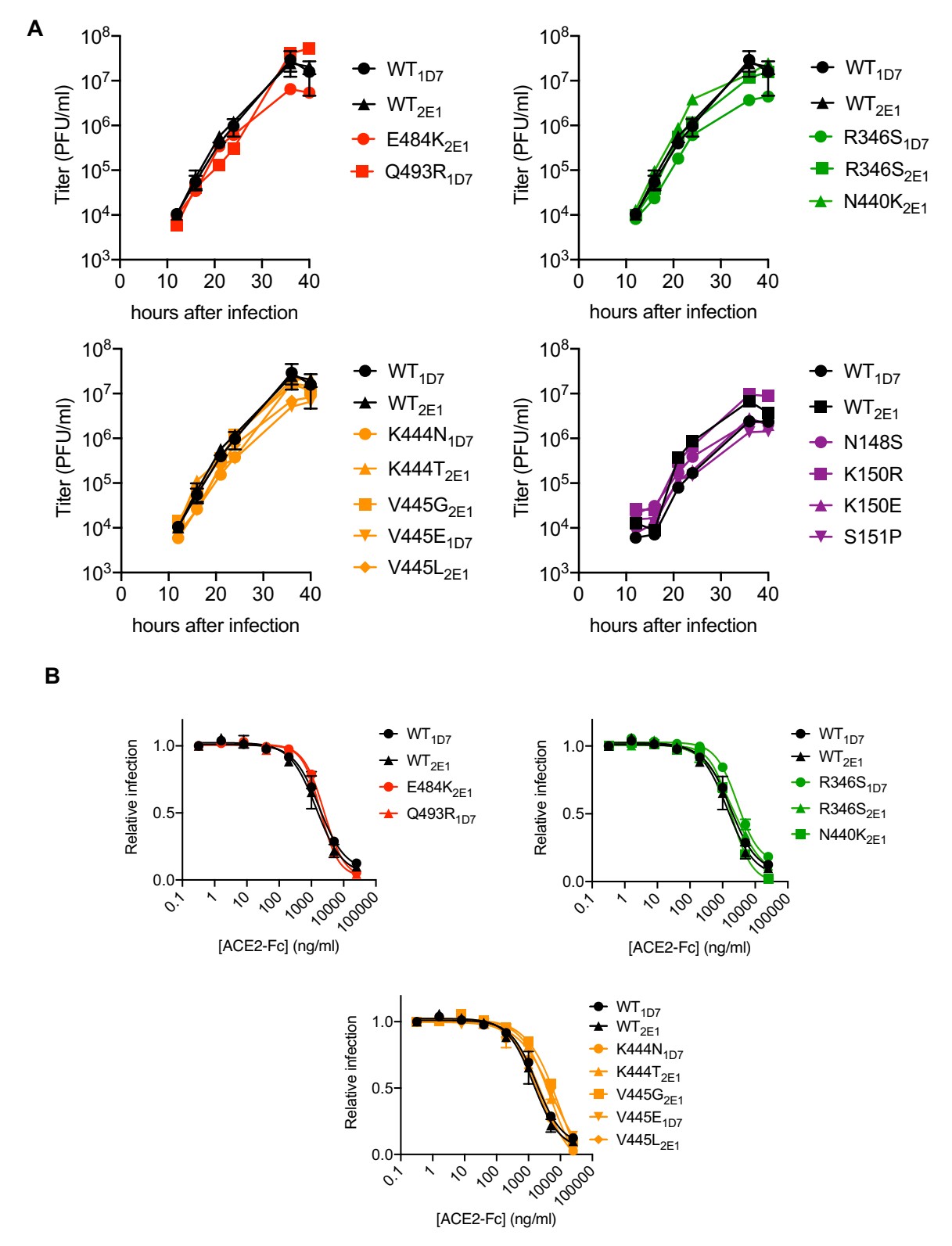

**Figure 3.** Characterization of mutant rVSV/SARS-CoV-2/GFP derivatives. (**A**) Replication of plaque-purified rVSV/SARS-CoV-2/GFP bearing individual S amino-acid substitutions that arose during passage with the indicated antibody or plasma. 293T/ACE2cl.22 cells were inoculated with equivalent doses of parental or mutant rVSV/SARS-CoV-2/GFP isolates. Supernatant was collected at the indicated times after inoculation and number of infectious units present therein was determined on 293T/ACE2cl.22 cells. The mean of two independent experiments is plotted. One set of WT controls run

*Figure 3 continued on next page*

*Figure 3 continued*

concurrently with the mutants are replotted in the upper and lower left panels, A different set of WT controls run concurrently with the mutants is shown in the lower right panel (**B**) Infection 293T/ACE2cl.22 cells by rVSV/SARS-CoV-2/GFP encoding the indicated S protein mutations in the presence of increasing amounts of a chimeric ACE2-Fc molecule. Infection was quantified by FACS. Mean of two independent experiments is plotted. The WT controls are replotted in each panel.

The online version of this article includes the following figure supplement(s) for figure 3:

**Figure supplement 1.** Example of plaque purification of individual viral mutants from populations passaged in the presence of antibodies.

(*Figure 4A,B*). The spatial distribution of these resistance-conferring mutations on the SARS-CoV-2 S RBD surface suggested the existence of both distinct and partly overlapping neutralizing epitopes on the RBD (*Figure 4C*). The C121 and C144 neutralizing epitopes appear to be similar, and include E484 and Q493, while a clearly distinct conformational epitope seems to be recognized by C135, that includes R346 and N440 residues. Antibodies that constitute at least part of the neutralizing activity evident in COV-NY plasma appear to recognize an epitope that includes and K444 and V445. The ability of mutations at these residues to confer partial resistance to C135 is consistent with their spatial proximity to the C135 conformational epitope (*Figure 4C*).

To test whether neutralization escape mutations conferred loss of binding to the monoclonal antibodies, we expressed conformationally prefusion-stabilized S-trimers (*Hsieh et al., 2020*), appended at their C-termini with NanoLuc luciferase (*Figure 5A*). The S-trimers were incubated in solution with the monoclonal antibodies, complexes were captured using protein G magnetic beads, and the amount of S-trimer captured was measured using NanoLuc luciferase assays (*Figure 5A*). As expected, C121, C135, and C144 monoclonal antibodies all bound the WT S-trimer (*Figure 5B*). The E484K and Q493R trimers exhibited complete, or near complete loss of binding to C121 and C144 antibodies but retained WT levels of binding to C135 (*Figure 5B*). Conversely, the R346S and N440K mutants exhibited complete loss of binding to C135, but retained WT levels of binding to C121 and C144. The K444N and V445E mutants retained near WT levels of binding to all three antibodies, despite exhibiting partial resistance to C135 (*Figure 5A,B*). Presumably the loss of affinity of these mutants for C135 was sufficient to impart partial neutralization resistance but insufficient to abolish binding in the solution binding assay.

Analysis of mutants that were isolated from the virus population that emerged during rVSV/SARS-CoV-2/GFP replication in the presence of COV-47 plasma (specifically N148S, K150R, K150E, S151P) revealed that these mutants exhibited specific resistance to COV-47 plasma. Indeed, the COV-47 plasma $NT_{50}$ for these mutants was reduced by 8- to 10-fold (*Figure 6A*). This finding indicates that the antibody or antibodies responsible for majority of neutralizing activity in COV-47 plasma target an NTD epitope that includes amino acids 148–151, even though the highly potent monoclonal antibody (C144) isolated from COV-47 targets the RBD. Mutants in the 148–151 NTD epitope exhibited marginal reductions in sensitivity to other plasmas (*Figure 6A*), indicating that different epitopes are primarily targeted by plasmas from the other donors.

The viral population that emerged during replication in COV-NY plasma yielded mutants K444N or T and V445G, E or L. Each of these mutations conferred substantial resistance to neutralization by COV-NY plasma, with ~ 10 to 30-fold reduction in $NT_{50}$ (*Figure 6B*). Thus, the dominant neutralizing activity in COV-NY plasma is represented by an antibody or antibodies recognizing an RBD epitope that includes K444 and V445. As was the case with COV-47 resistant mutants, viruses encoding the mutations conferring resistance to COV-NY plasma retained almost full sensitivity to neutralization by other plasmas (*Figure 6B*).

Interestingly, the mutations that conferred complete or near complete resistance to the potent RBD-specific monoclonal antibodies C144, C135, and C121 conferred little or no resistance to neutralization by plasma from the same individual, or other individuals (*Figure 6C*). These RBD-specific antibodies represent the most potent monoclonal antibodies isolated from COV-47, COV-72, and COV-107, respectively, but the retention of plasma sensitivity by the monoclonal antibody-resistant mutants suggests that these antibodies contribute little to the overall neutralization activity of plasma from the same individual. This finding is consistent with the observation that memory B cells producing these antibodies are rare (*Robbiani et al., 2020*), and with the results of the selection experiments in which rVSV/SARS-CoV-2/GFP replication in the presence of COV-47, COV-72, and

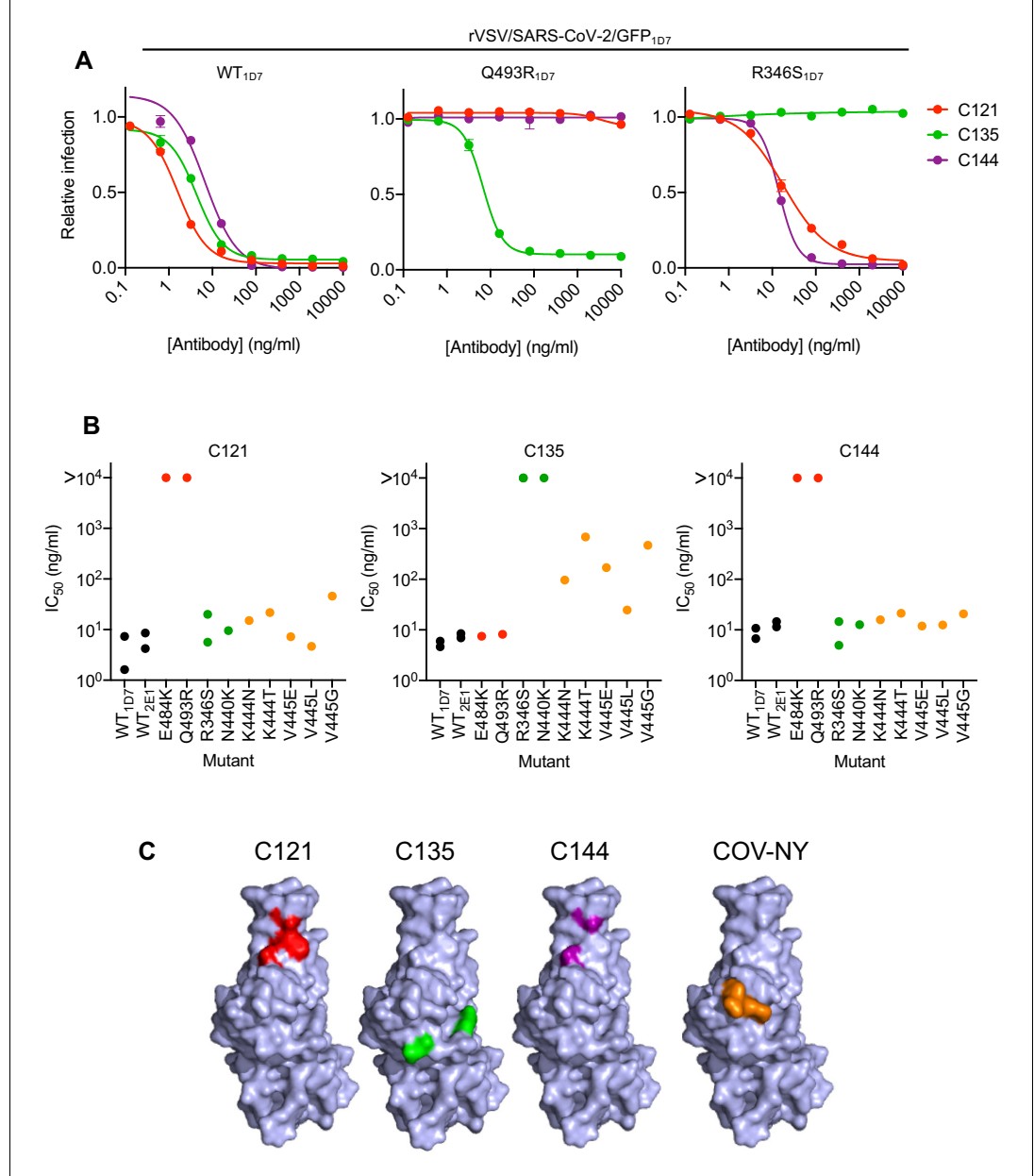

**Figure 4.** Neutralization of rVSV/SARS-CoV-2/GFP RBD mutants by monoclonal antibodies. (**A**) Examples of neutralization resistance of rVSV/SARS-CoV-2/GFP mutants that were isolated following passage in the presence of antibodies. 293T/ACE2cl.22 cells were inoculated with WT or mutant rVSV/SARS-CoV-2/GFP in the presence of increasing amount of each monoclonal antibody, and infection quantified by FACS 16 hr later. Mean and SD from two technical replicates, representative of two independent experiments. (**B**) Neutralization sensitivity/resistance of rVSV/SARS-CoV-2/GFP mutants isolated following replication in the presence of antibodies. Mean $IC_{50}$ values were calculated for each virus-monoclonal antibody combination in two independent experiments. (**C**) Position of neutralization resistance-conferring substitutions. Structure of the RBD (from PDB 6M17 *Yan et al., 2020*) with positions that are occupied by amino acids where mutations were acquired during replication in the presence of each monoclonal antibody or COV-NY plasma indicated.

COV-107 plasma did not enrich for mutations that correspond to the neutralization epitopes targeted by the monoclonal antibodies obtained from these individuals (*Figure 2—figure supplement 1A,B*, *Figure 2—figure supplement 2A,B*. *Table 2*). Overall, analysis of even this limited set of monoclonal antibodies and plasmas shows that potent neutralization can be conferred by antibodies that target diverse SARS-CoV-2 epitopes. Moreover, the most potently neutralizing antibodies

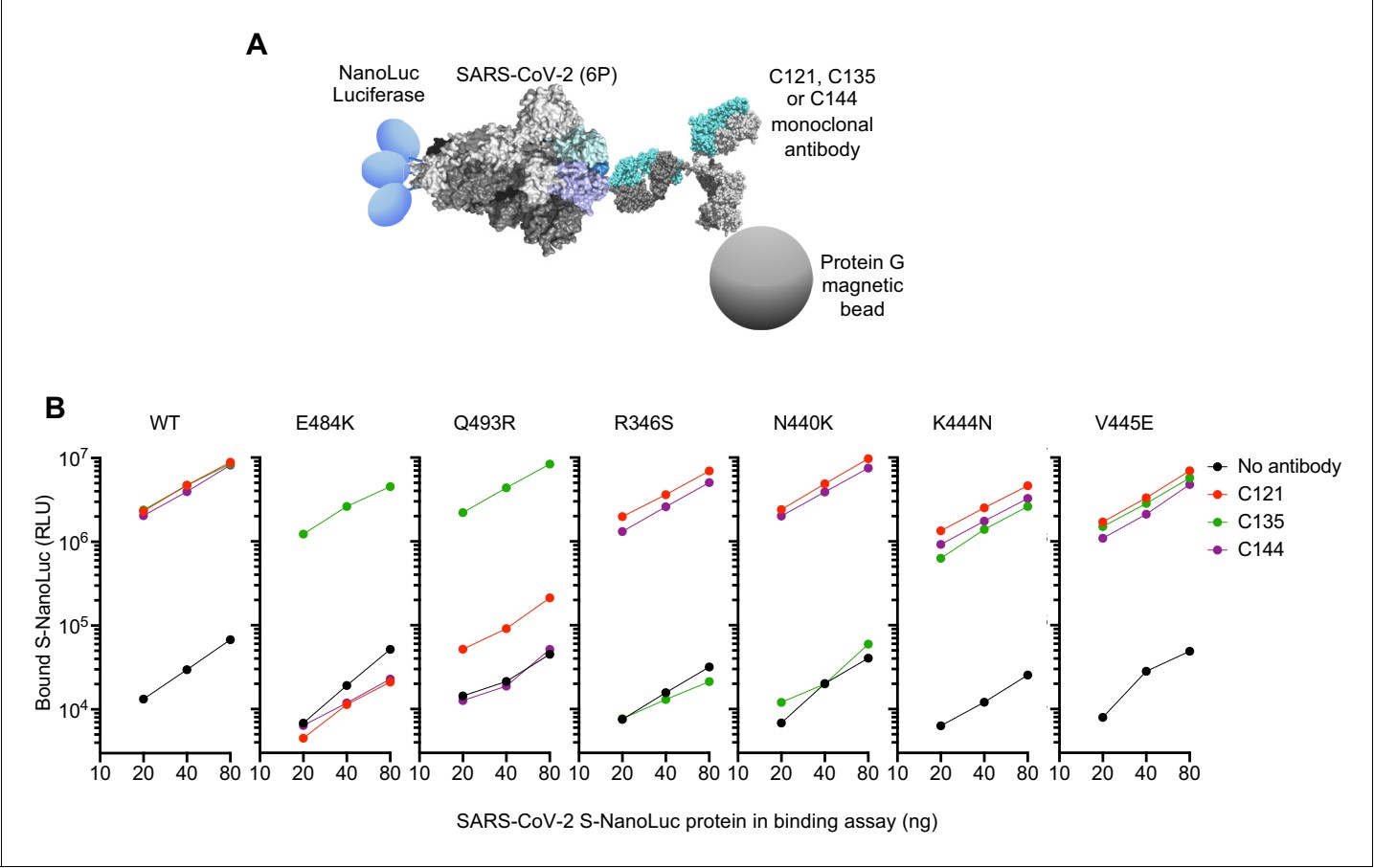

**Figure 5.** Loss of binding to monoclonal antibodies by neutralization escape mutants. (A) Schematic representation of the binding assay in which NanoLuc luciferase is appended to the C-termini of a conformationally stabilized S-trimer. The fusion protein is incubated with antibodies and complexes captured using protein G magnetic beads (B) Bound Nanoluc luciferase quantified following incubation of the indicated WT or mutant Nanoluc-S fusion proteins with the indicated antibodies and Protein G magnetic beads. Mean of three technical replicates at each S-Nanoluc concentration.

generated in a given COVID19 convalescent individual may contribute in only a minor way to the overall neutralizing antibody response in that same individual (see discussion).

## Natural occurrence of antibody-resistance RBD mutations

The aforementioned neutralizing antibody escape mutations were artificially generated during in vitro replication of a recombinant virus. However, as monoclonal antibodies are developed for therapeutic and prophylactic applications, and vaccine candidates are deployed, and the possibility of SARS-CoV-2 reinfection becomes greater, it is important both to understand pathways of antibody resistance and to monitor the prevalence of resistance-conferring mutations in naturally circulating SARS-CoV-2 populations.

To survey the natural occurrence of mutations that might confer resistance to the monoclonal and plasma antibodies used in our experiments we used the GISAID (*Elbe et al., 2017*) and CoV-Glue (*Singer et al., 2020*) SARS-CoV-2 databases. Among the 55,189 SARS-CoV-2 sequences in the CoV2-Glue database at the time of writing, 2175 different non-synonymous mutations were present in natural populations of SARS-CoV-2 S protein sequences. Consistent with the finding that none of the mutations that arose in our selection experiments gave an obvious fitness deficit (in the context of rVSV/SARS-CoV-2/GFP), most were also present in natural viral populations.

For phenotypic analysis of naturally occurring SARS-CoV-2 S mutations, we focused on the ACE2 interface of the RBD, as it is the target of most therapeutic antibodies entering clinical development (*Robbiani et al., 2020*; *Hansen et al., 2020*; *Baum et al., 2020*), and is also the target of at least

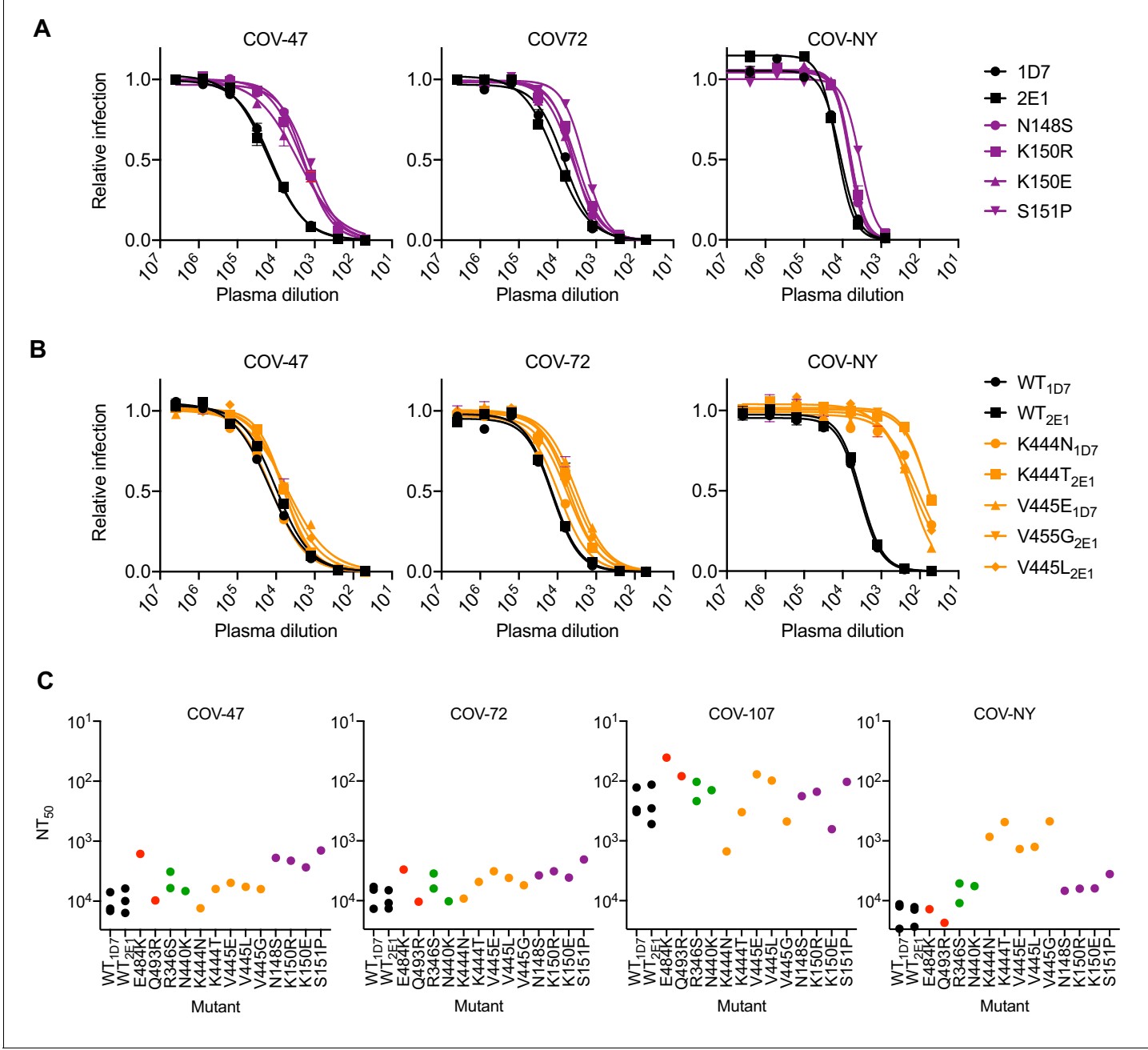

**Figure 6.** Neutralization of rVSV/SARS-CoV-2/GFP RBD mutants by convalescent plasma. (A, B) Neutralization of rVSV/SARS-CoV-2/GFP mutants isolated following replication in the presence COV-47 plasma (A) or COV-NY plasma (B). 293T/ACE2cl.22 cells were inoculated with WT or mutant rVSV/ SARS-CoV-2/GFP in the presence of increasing amounts of the indicated plasma, and infection quantified by flow cytometry, 16 hr later. Mean of two technical replicates, representative of two independent experiments (C) Plasma neutralization sensitivity/resistance of rVSV/SARS-CoV-2/GFP mutants isolated following replication in the presence of monoclonal antibodies or convalescent plasma. Mean $NT_{50}$ values were calculated for each virus-plasma combination from two independent experiments.

some antibodies present in convalescent plasmas. In addition to the mutations that arose in our antibody selection experiments, inspection of circulating RBD sequences revealed numerous naturally occurring mutations in the vicinity of the ACE2 binding site and the epitopes targeted by the antibodies (https://www.gisaid.org, http://cov-glue.cvr.gla.ac.uk) (*Figure 7A–D*). We tested nearly all of the mutations that are present in the GISAID database as of June 2020, in the proximity of the ACE2 binding site and neutralizing epitopes, for their ability to confer resistance to the monoclonal

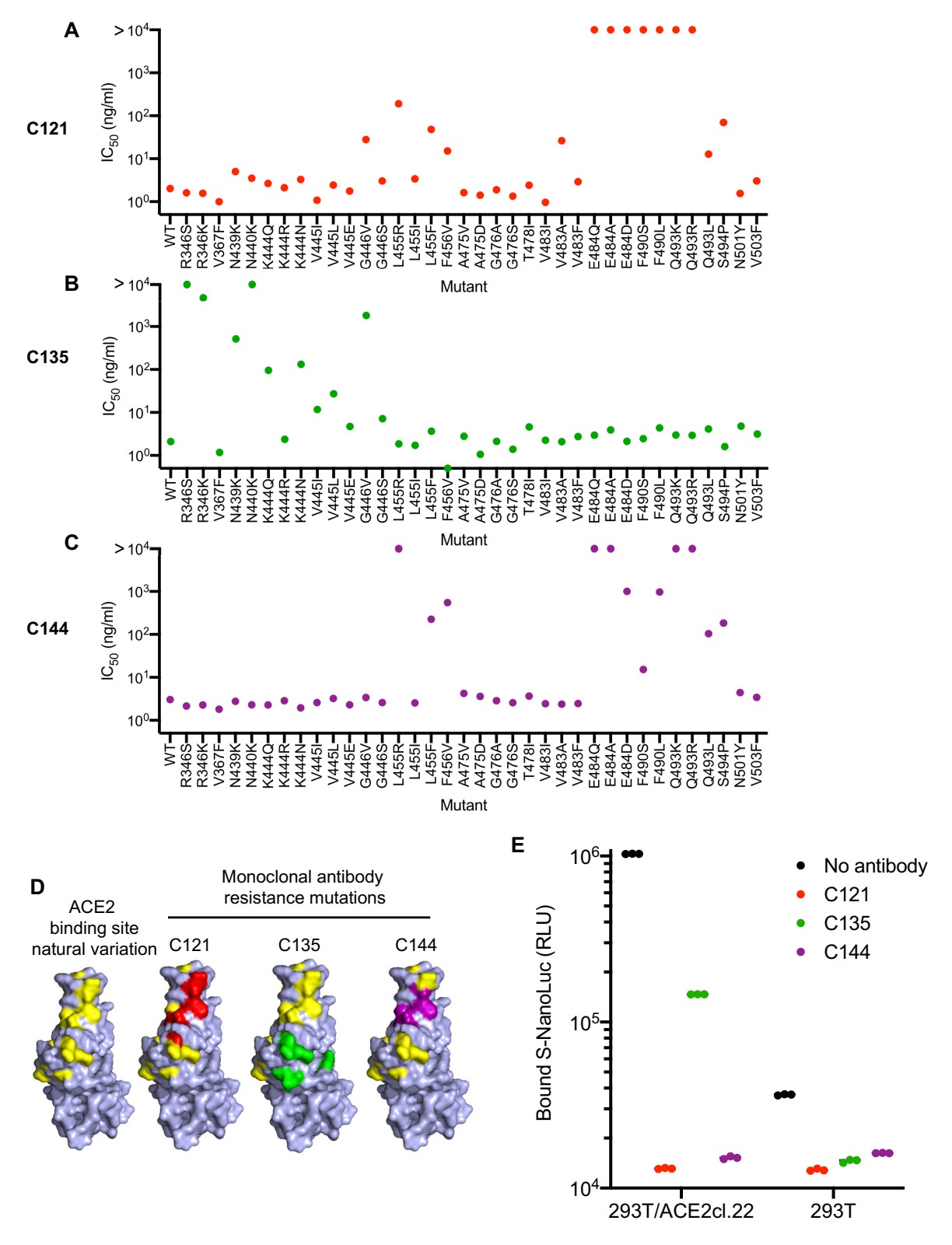

**Figure 7.** Effects of naturally occurring RBD amino-acid substitutions on S sensitivity to neutralizing monoclonal antibodies. (A–C) Neutralization of HIV-based reporter viruses pseudotyped with SARS-CoV-2 S proteins harboring the indicated naturally occurring substitutions. 293T/ACE2cl.22 cells were inoculated with equivalent doses of each pseudotyped virus in the presence of increasing amount of C121 (A) C135 (B) or C144 (C). Mean $IC_{50}$ values were calculated for each virus-antibody combination from two independent experiments. (D) Position of substitutions conferring neutralization

*Figure 7 continued on next page*

*Figure 7 continued*

resistance relative to the amino acids close to the ACE2 binding site whose identity varies in global SARS-CoV-2 sequences. The RBD structure (from PDB 6M17 *Yan et al., 2020*) is depicted with naturally varying amino acids close to the ACE2 binding site colored in yellow. Amino acids whose substitution confers partial or complete (IC50 > 10 µg/ml) resistance to each monoclonal antibody in the HIV-pseudotype assays are indicated for C121 (red) C135 (green) and C144 (purple). (E) Binding of S-NanoLuc fusion protein in relative light units (RLU) to 293T or 293T/ACE2cl.22 cells after preincubation in the absence or presence of C121, C135, and C144 monoclonal antibodies. Each symbol represents a technical replicate.

antibodies, using an HIV-1-based pseudotyped virus-based assay (*Figure 7A–C*). Consistent with, and extending our findings with rVSV/SARS-CoV-2/GFP, naturally occurring mutations at positions E484, F490, Q493, and S494 conferred complete or partial resistance to C121 and C144 (*Figure 7A, C*). While there was substantial overlap in the mutations that caused resistance to C121 and C144, there were also clear differences in the degree to which certain mutations (e.g. G446, L455R/I/F, F490S/L) affected sensitivity to the two antibodies. Naturally occurring mutations that conferred complete or partial resistance to C135 were at positions R346, N439, N440, K444, V445 and G446. In contrast to the C121/C144 epitope, these amino acids are peripheral to the ACE2 binding site on the RBD (*Figure 7D*). Indeed, in experiments where the binding of a conformationally stabilized tri-meric S-NanoLuc fusion protein to 293T/ACE2cl.22 cells was measured, preincubation of S-NanoLuc with a molar excess of C121 or C144 completely blocked binding (*Figure 7E*). Conversely, preincu-bation with C135 only partly blocked binding to 293T/ACE2cl.22 cells, consistent with the finding that the C135 conformational epitope does not overlap the ACE2 binding site (*Figure 7D*). C135 might inhibit S-ACE-2 binding by steric interference with access to the ACE two binding site. These results are also consistent with experiments which indicated that C135 does not compete with C121 and C144 for binding to the RBD (*Robbiani et al., 2020*).

All of the mutations that were selected in our rVSV/SARS-CoV-2/GFP antibody selection experiments as well as other mutations that confer resistance to C121, C144, or C135 are found in naturally circulating SARS-CoV-2 populations at very low frequencies (*Figure 8*). With one exception (N439K) that is circulating nearly exclusively in Scotland and is present in ~ 1% of COV-Glue database sequences, (and whose frequency may be overestimated due to regional oversampling) all antibody resistance mutations uncovered herein are present in global SARS-CoV-2 at frequencies of < 1 in 1000 sequences (*Figure 8*). The frequency with which the resistance mutations are present in

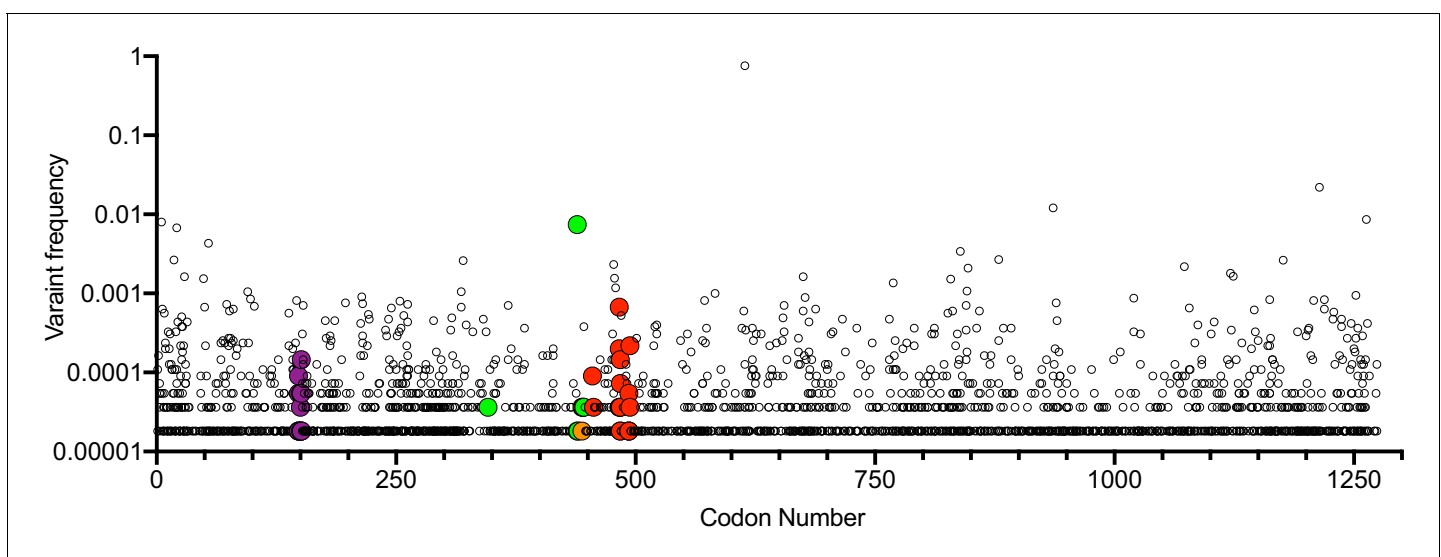

**Figure 8.** Position and frequency of S amino-acid substitutions in SARS-CoV-2 S. Global variant frequency reported by CoV-Glue in the SARS-CoV-2 S protein. Each individual variant is indicated by a symbol whose position in the S sequence is indicated on the X-axis and frequency reported by COV-Glue is indicated on the Y-axis. Individual substitutions at positions where mutations conferring resistance to neutralizing antibodies or plasma were found herein are indicated by enlarged and colored symbols: red for C121 and C144, green for C135, purple for COV-47 plasma and orange for COV-NY plasma. The common D/G614 variant is indicated.

naturally occurring SARS-CoV-2 sequences appeared rather typical compared to other S mutations, with the caveat that sampling of global SARS-CoV-2 is nonrandom. Therefore, these observations do not provide evidence that the neutralizing activities exhibited by the monoclonal antibodies or plasma samples used herein have driven strong selection of naturally circulating SARS-CoV-2 sequences thus far (*Figure 8*).

## Selection of combinations of monoclonal antibodies for therapeutic and prophylactic applications

The ability of SARS-CoV-2 monoclonal antibodies and plasma to select variants that are apparently fit and that naturally occur at low frequencies in circulating viral populations suggests that therapeutic use of single antibodies might select for escape mutants. To mitigate against the emergence or selection of escape mutations during therapy, or during population-based prophylaxis, we tested whether combinations of monoclonal antibodies could suppress the emergence of resistant variants during in vitro selection experiments. Specifically, we repeated antibody selection experiments in which rVSV/SARS-CoV-2/GFP populations containing $10^6$ infectious virions were incubated with 10 µg/ml of each individual monoclonal antibody, or mixtures containing 5 µg/ml of each of two antibodies (*Figure 9*). C121 and C144 target largely overlapping epitopes, and mutations conferring resistance to one of these antibodies generally conferred resistance to the other (*Figure 7A–D*). Therefore, we used mixtures of antibodies targeting clearly distinct epitopes (C121+C135 and C144+C135). As previously, replication of rVSV/SARS-CoV-2/GFP in the presence of a single monoclonal antibody enabled the formation of infected foci in p1 cultures (*Figure 9A–C*), that rapidly expanded and enabled the emergence of apparently resistant virus populations. Indeed, rVSV/SARS-CoV-2/GFP yields from p2 cultures established with one antibody (C121, C135 or C144) were indistinguishable from those established with no antibody (*Figure 9D*). Conversely, rVSV/SARS-CoV-2/GFP replication in the presence of mixtures of C121+C135 or C144+C135 led to sparse infection of individual cells in p1 cultures, but there was little or no formation of foci that would suggest propagation of infection from these infected cells (*Figure 9A–C*), Therefore, it is likely that infected cells arose from rare, non-neutralized, virions that retained sensitivity to at least one of the antibodies in mixture. Consequently, viral spread was apparently completely suppressed and no replication-competent rVSV/SARS-CoV-2/GFP was detected in p2 cultures established with mixtures of the two antibodies (*Figure 9D*).

## Discussion

The degree to which resistance will impact effectiveness of antibodies in SARS-CoV-2 therapeutic and vaccine settings is currently unclear (*Baum et al., 2020*). Notably, the inter-individual variation in SARS-CoV-2 sequences is low compared to many other RNA viruses (*van Dorp et al., 2020*; *Rambaut et al., 2020*; *Dearlove et al., 2020*; *Rausch et al., 2020*), in part because coronaviruses encode a 3′−5′ exonuclease activity. The exonuclease activity provides a proofreading function that enhances replication fidelity and limits viral sequence diversification (*Denison et al., 2011*).

However, replication fidelity is but one of several variables that affect viral population diversity (*Duffy et al., 2008*; *Moya et al., 2000*). One determinant of total viral diversity is population size. Many millions of individuals have been infected by SARS-CoV-2, and a single swab from an infected individual can contain in excess of $10^9$ copies of viral RNA (*Wölfel et al., 2020*). It follows that SARS-CoV-2 genomes encoding every possible single amino-acid substitution are present in the global population, and perhaps in a significant fraction of individual COVID19 patients. Thus, the frequency with which particular variants occur in the global SARS-CoV-2 population is strongly influenced by the frequency with which negative and positive selection pressures that favor their propagation are encountered, as well as founder effects at the individual patient and population levels (*Korber et al., 2020*).

Fitness effects of mutations will obviously vary and will suppress the prevalence of deleterious mutations (*Dolan et al., 2018*; *Andino and Domingo, 2015*). However, otherwise neutral, or even modestly deleterious, mutations will rise in prevalence if they confer escape from selective pressures, such as immune responses. The prevalence of neutralizing antibody escape mutations will also be strongly influenced by the frequency with which SARS-CoV-2 encounters neutralizing antibodies. Peak viral burden in swabs and sputum, which likely corresponds to peak infectiousness and

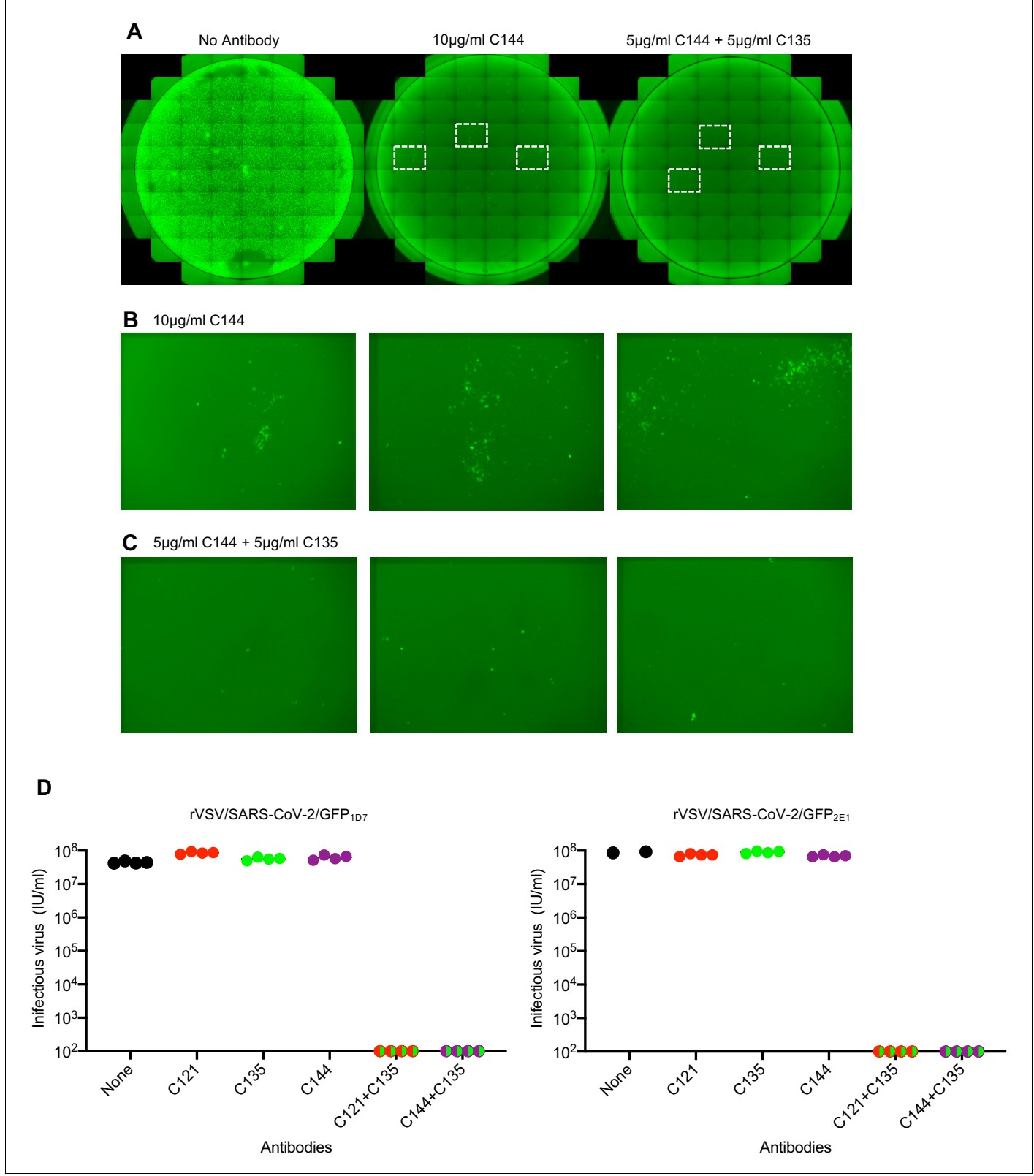

**Figure 9.** Suppression of antibody resistance through the use of antibody combinations. (A) Representative images of 293T/ACE2 (B) cells infected with the equivalent doses of rVSV/SARS-CoV-2/GFP in the absence or presence of 10 µg/ml of one (C144) or 5 µg/ml of each of two (C144 +C135) neutralizing monoclonal antibodies. (B) Expanded view of the boxed areas containing individual plaques from the culture infected in the presence of 10 µg/ml C144. (C) Expanded view of the boxed areas in A containing infected cells from the culture infected in the presence of 5 µg/ml each of (C144

*Figure 9 continued on next page*

*Figure 9 continued*

and C135). (D) Infectious virus yield following two passages of rVSV/SARS-CoV-2/GFP in the absence or presence of individual neutralizing antibodies or combinations of two antibodies. Titers were determined on 293T/ACE2cl.22 cells. Each symbol represents a technical replicate and results from two independent experiments using rVSV/SARS-CoV-2/GFP$_{1D7}$ and rVSV/SARS-CoV-2/GFP$_{2E1}$ are shown.

frequency of transmission events, appears to approximately correspond with the onset of symptoms, and clearly occurs before seroconversion (*Wölfel et al., 2020*). Thus, it is quite plausible that most transmission events involve virus populations that are yet to experience antibody-imposed selective pressure in the transmitting individual. Such a scenario would reduce the occurrence of antibody escape mutations in natural viral populations. It will be interesting to determine whether viral sequences obtained late in infection are more diverse or have evidence of immunological escape mutations.

There are situations that are anticipated to increase the frequency of encounters between SARS-CoV-2 and antibodies that could impact the emergence of antibody resistance. Millions of individuals have already been infected with SARS-CoV-2 and among them, neutralizing antibody titers are extremely variable (*Robbiani et al., 2020*; *Wu et al., 2020b*; *Luchsinger et al., 2020*). Those with weak immune responses or waning immunity could become re-infected, and if so, that encounters between SARS-CoV-2 and pre-existing but incompletely protective neutralizing antibodies might drive the selection of escape variants (*Kk et al., 2020a*; *Van Elslande et al., 2020*; *Larson et al., 2020*; *Kk et al., 2020b*). In a similar manner, poorly immunogenic vaccine candidates, convalescent plasma therapy, and suboptimal monoclonal antibody treatment, particularly monotherapy (*Baum et al., 2020*), could create conditions to drive the acquisition of resistance to commonly occurring antibodies in circulating virus populations.

The extent to which SARS-CoV-2 evasion of individual antibody responses would have pervasive effects on the efficacy of vaccines and monoclonal antibody treatment/therapy will also be influenced by the diversity of neutralizing antibody responses within and between individuals. Analysis of potent neutralizing antibodies cloned by several groups indicates that potent neutralizing antibodies are commonly elicited, and very similar antibodies, such as those containing IGHV3-53 and IGHV3-66 can be found in different individuals (*Robbiani et al., 2020*; *Barnes et al., 2020*; *Yuan et al., 2020*). These findings imply a degree of homogeneity the among neutralizing antibodies that are generated in different individuals. Nevertheless, each of the four convalescent plasma tested herein had distinct neutralizing characteristics. In two of the four plasma tested, selection experiments suggested that a dominant antibody specificity was responsible for a significant fraction of the neutralizing capacity of the plasma. However, the failure of single amino-acid substitutions to confer complete resistance to any plasma strongly suggests the existence of multiple neutralizing specificities in each donor. Indeed, in one example (COV47), viral mutants that were completely resistant to a potent monoclonal antibody from that donor (C144), retained near complete sensitivity to plasma from that same individual, Thus, in that individual other antibodies in the plasma, not the most potent monoclonal antibody, must dominate the neutralizing activity of the plasma. That COV47 plasma selected mutations at a different site in S (NTD) to that selected by C144 (RBD), is orthogonal supportive evidence that the C144 monoclonal antibody does not constitute the major neutralizing activity in the plasma of COV47.

The techniques described herein could be adapted to broadly survey the diversity of SARS-CoV-2 neutralizing specificities in many plasma samples following natural infection or vaccination, enabling a more complete picture of the diversity of SARS-CoV-2 neutralizing antibody responses to be developed. Indeed, the approach described herein can be used to map epitopes of potent neutralizing antibodies rapidly and precisely. It has an advantage over other epitope mapping approaches (such as array-based oligo-peptide scanning or random site-directed mutagenesis) (*Greaney et al., 2020*), in that selective pressure acts solely on the naturally formed, fusion-competent viral spike. While mutations outside the antibody binding sites might lead to resistance, the functional requirement will prohibit mutations that simply disrupt the native conformation. Indeed, we found that the neutralizing antibody escape mutations described herein did not detectably alter rVSV/SARS-CoV-2/GFP replication, did not affect ACE2-Fc sensitivity and were found in natural populations at unexceptional frequencies, consistent with the notion that they do not have large effects on fitness in the absence of neutralizing antibodies. That said, the selection/viral evolution scheme described herein

approximates to but does not precisely recapitulate the evolutionary dynamics that would play out in natural SARS-CoV-2 infection. Differences in viral populations sizes, replication fidelity and fitness effects of mutations in SARS-CoV-2 versus rVSV/SARS-CoV-2/GFP, and the complexity and changing nature of evolving antibody responses could all affect the nature of, and response to, selection pressures. For example, one caveat is the utilization of plasma for selection experiments. Immunoglobulin subtypes (e.g. IgA versus IgG) are differentially represented in plasma versus the respiratory tract, and the concentrations of each immunoglobulin subtype or specificity precisely at the sites of SARS-CoV-2 replication is unknown. However, it is known that IgG that dominates plasma immunoglobulins is also present in lung secretions, albeit at lower levels than IgA (*Burnett, 1986*). Moreover, our recent work has indicated that at least some of the neutralizing activity in plasma is contributed by IgA and several of the antibody lineages that we have cloned from SARS-CoV-2 convalescents are class switched to both IgA and IgG (*Wang et al., 2020*). Overall, it is likely that the antibody specificities present in the respiratory tract broadly reflect, but perhaps do not precisely recapitulate, those present in the plasma used for the selection experiments described herein.

Human monoclonal antibodies targeting both the NTD and RBD of SARS-CoV-2 have been isolated, with those targeting RBD being especially potent. As these antibodies are used clinically (*Hansen et al., 2020*; *Baum et al., 2020*), in therapeutic and prophylactic modes, it will be important to identify resistance mutations and monitor their prevalence in a way that is analogous to antiviral and antibiotic resistance monitoring in other infectious diseases. Moreover, as is shown herein, the selection of antibody mixtures with non-overlapping escape mutations should reduce the emergence of resistance and prolong the utility of antibody therapies in SARS-CoV-2 infection.

# Materials and methods

**Key resources table**

| Reagent type (species) or resource | Designation | Source or reference | Identifiers | Additional information |
|---|---|---|---|---|
| Strain, strain background (Vesicular Stomatitis Virus) | VSV/SARS-CoV-2/GFP$_{1D7}$; WT$_{1D7}$ | *Schmidt et al., 2020* | | Recombinant chimeric VSV/SARS-CoV-2 reporter virus |
| Strain, strain background (Vesicular Stomatitis Virus) | rVSV/SARS-CoV-2/GFP$_{2E1}$; WT$_{2E1}$ | *Schmidt et al., 2020* | | Recombinant chimeric VSV/SARS-CoV-2 reporter virus |
| Strain, strain background (Vesicular Stomatitis Virus) | E484K$_{2E1}$ | *Schmidt et al., 2020*, and this paper | | mutant rVSV/SARS-CoV-2/GFP derivative Inquiries should be addressed to P.Bieniasz |
| Strain, strain background (Vesicular Stomatitis Virus) | Q493R$_{1D7}$ | *Schmidt et al., 2020*, and this paper | | mutant rVSV/SARS-CoV-2/GFP derivative Inquiries should be addressed to P.Bieniasz |
| Strain, strain background (Vesicular Stomatitis Virus) | R346S$_{1D7}$ | *Schmidt et al., 2020*, and this paper | | mutant rVSV/SARS-CoV-2/GFP derivative Inquiries should be addressed to P.Bieniasz |
| Strain, strain background (Vesicular Stomatitis Virus) | R346$_{2E1}$ | *Schmidt et al., 2020*, and this paper | | mutant rVSV/SARS-CoV-2/GFP derivative Inquiries should be addressed to P.Bieniasz |
| Strain, strain background (Vesicular Stomatitis Virus) | N440K$_{2E1}$ | *Schmidt et al., 2020*, and this paper | | mutant rVSV/SARS-CoV-2/GFP derivative Inquiries should be addressed to P.Bieniasz |

*Continued on next page*

*Continued*

| Reagent type (species) or resource | Designation | Source or reference | Identifiers | Additional information |
|---|---|---|---|---|
| Strain, strain background (Vesicular Stomatitis Virus) | K444N$_{1D7}$ | *Schmidt et al., 2020*, and this paper | | mutant rVSV/SARS-CoV-2/GFP derivative Inquiries should be addressed to P.Bieniasz |
| Strain, strain background (Vesicular Stomatitis Virus) | K444T$_{2E1}$ | *Schmidt et al., 2020*, and this paper | | mutant rVSV/SARS-CoV-2/GFP derivative Inquiries should be addressed to P.Bieniasz |
| Strain, strain background (Vesicular Stomatitis Virus) | V445G$_{2E1}$ | *Schmidt et al., 2020*, and this paper | | mutant rVSV/SARS-CoV-2/GFP derivative Inquiries should be addressed to P.Bieniasz |
| Strain, strain background (Vesicular Stomatitis Virus) | V445E$_{1D7}$ | *Schmidt et al., 2020*, and this paper | | mutant rVSV/SARS-CoV-2/GFP derivative Inquiries should be addressed to P.Bieniasz |
| Strain, strain background (Vesicular Stomatitis Virus) | V445L$_{2E1}$ | *Schmidt et al., 2020*, and this paper | | mutant rVSV/SARS-CoV-2/GFP derivative Inquiries should be addressed to P.Bieniasz |
| Strain, strain background (Vesicular Stomatitis Virus) | N148S | *Schmidt et al., 2020*, and this paper | | mutant rVSV/SARS-CoV-2/GFP derivative Inquiries should be addressed to P.Bieniasz |
| Strain, strain background (Vesicular Stomatitis Virus) | K150R | *Schmidt et al., 2020*, and this paper | | mutant rVSV/SARS-CoV-2/GFP derivative Inquiries should be addressed to P.Bieniasz |
| Strain, strain background (Vesicular Stomatitis Virus) | K150E | *Schmidt et al., 2020*, and this paper | | mutant rVSV/SARS-CoV-2/GFP derivative Inquiries should be addressed to P.Bieniasz |
| Strain, strain background (Vesicular Stomatitis Virus) | S151P | *Schmidt et al., 2020*, and this paper | | mutant rVSV/SARS-CoV-2/GFP derivative Inquiries should be addressed to P.Bieniasz |
| Cell line (*Homo sapiens*) | Expi293F Cells | Thermo Fisher Scientific | Cat# A14527 | |
| Cell line (*H. sapiens*) | 293T (embryonic, kidney) | ATCC | CRL-3216 | |
| Cell line (*H. sapiens*) | 293T/ACE2(B) | *Schmidt et al., 2020* | | 293 T cells expressing human ACE2 (bulk population) |
| Cell line (*H. sapiens*) | 293T/ACE2cl.22 | *Schmidt et al., 2020* | | 293 T cells expressing human ACE2 (single cell clone) |
| Biological sample (*H. sapiens*) | COV-47 | *Robbiani et al., 2020* | | Human plasma sample |
| Biological sample (*H. sapiens*) | COV-72 | *Robbiani et al., 2020* | | Human plasma sample |
| Biological sample (*H. sapiens*) | COV-107 | *Robbiani et al., 2020* | | Human plasma sample |

*Continued on next page*

*Continued*

| Reagent type (species) or resource | Designation | Source or reference | Identifiers | Additional information |
|---|---|---|---|---|
| Biological sample (*H. sapiens*) | COV-NY | *Luchsinger et al., 2020* | | Human plasma sample |
| Antibody | C121 (Human monoclonal) | *Robbiani et al., 2020* | | Selection experiments (10 µg/ml, 5 µg/ml) |
| Antibody | C135 (Human monoclonal) | *Robbiani et al., 2020* | | Selection experiments (10 µg/ml, 5 µg/ml) |
| Antibody | C144 (Human monoclonal) | *Robbiani et al., 2020* | | Selection experiments (10 µg/ml, 5 µg/ml) |
| Recombinant DNA reagent | CSIB(ACE2) | *Schmidt et al., 2020* | | |
| Recombinant DNA reagent | pHIV$_{NL}$GagPol | *Schmidt et al., 2020* | | |
| Recombinant DNA reagent | pCCNano Luc2AEGFP | *Schmidt et al., 2020* | | |
| Recombinant DNA reagent | pSARS-CoV-2Δ19 | *Schmidt et al., 2020* | | Epression plasmid containing a C-terminally truncated SARS-CoV-2 S protein (pSARS-CoV-2Δ19) containing a synthetic human-codon-optimized cDNA (Geneart) |
| Recombinant DNA reagent | R346S | *Schmidt et al., 2020*, and this paper | | pSARS-CoV-2Δ19 containing the indicated mutation. Inquiries should be addressed to P.Bieniasz |
| Recombinant DNA reagent | R346K | *Schmidt et al., 2020*, and this paper | | pSARS-CoV-2Δ19 containing the indicated mutation. Inquiries should be addressed to P.Bieniasz |
| Recombinant DNA reagent | V367F | *Schmidt et al., 2020*, and this paper | | pSARS-CoV-2Δ19 containing the indicated mutation. Inquiries should be addressed to P.Bieniasz |
| Recombinant DNA reagent | N439K | *Schmidt et al., 2020*, and this paper | | pSARS-CoV-2Δ19 containing the indicated mutation. Inquiries should be addressed to P.Bieniasz |
| rRcombinant DNA reagent | N440K | *Schmidt et al., 2020*, and this paper | | pSARS-CoV-2Δ19 containing the indicated mutation. Inquiries should be addressed to P.Bieniasz |
| Recombinant DNA reagent | K444Q | *Schmidt et al., 2020*, and this paper | | pSARS-CoV-2Δ19 containing the indicated mutation. Inquiries should be addressed to P.Bieniasz |

*Continued*

| Reagent type (species) or resource | Designation | Source or reference | Identifiers | Additional information |
|---|---|---|---|---|
| Recombinant DNA reagent | K444R | *Schmidt et al., 2020*, and this paper | | pSARS-CoV-2Δ19 containing the indicated mutation. Inquiries should be addressed to P.Bieniasz |
| Recombinant DNA reagent | K444N | *Schmidt et al., 2020*, and this paper | | pSARS-CoV-2Δ19 containing the indicated mutation. Inquiries should be addressed to P.Bieniasz |
| Recombinant DNA reagent | V445I | *Schmidt et al., 2020*, and this paper | | pSARS-CoV-2Δ19 containing the indicated mutation. Inquiries should be addressed to P.Bieniasz |
| Recombinant DNA reagent | V445E | *Schmidt et al., 2020*, and this paper | | pSARS-CoV-2Δ19 containing the indicated mutation. Inquiries should be addressed to P.Bieniasz |
| Recombinant DNA reagent | V445L | *Schmidt et al., 2020*, and this paper | | pSARS-CoV-2Δ19 containing the indicated mutation. Inquiries should be addressed to P.Bieniasz |
| Recombinant DNA reagent | V445K | *Schmidt et al., 2020*, and this paper | | pSARS-CoV-2Δ19 containing the indicated mutation. Inquiries should be addressed to P.Bieniasz |
| Recombinant DNA reagent | G446V | *Schmidt et al., 2020*, and this paper | | pSARS-CoV-2Δ19 containing the indicated mutation. Inquiries should be addressed to P.Bieniasz |
| Recombinant DNA reagent | G446S | *Schmidt et al., 2020*, and this paper | | pSARS-CoV-2Δ19 containing the indicated mutation. Inquiries should be addressed to P.Bieniasz |
| Recombinant DNA reagent | L455R | *Schmidt et al., 2020*, and this paper | | pSARS-CoV-2Δ19 containing the indicated mutation. Inquiries should be addressed to P.Bieniasz |
| Recombinant DNA reagent | L455I | *Schmidt et al., 2020*, and this paper | | pSARS-CoV-2Δ19 containing the indicated mutation. Inquiries should be addressed to P.Bieniasz |
| Recombinant DNA reagent | L455F | *Schmidt et al., 2020*, and this paper | | pSARS-CoV-2Δ19 containing the indicated mutation. Inquiries should be addressed to P.Bieniasz |
| Recombinant DNA reagent | F456V | *Schmidt et al., 2020*, and this paper | | pSARS-CoV-2Δ19 containing the indicated mutation. Inquiries should be addressed to P.Bieniasz |

*Continued on next page*

Continued

| Reagent type (species) or resource | Designation | Source or reference | Identifiers | Additional information |
|---|---|---|---|---|
| Recombinant DNA reagent | A475V | *Schmidt et al., 2020*, and this paper | | pSARS-CoV-2Δ19 containing the indicated mutation. Inquiries should be addressed to P.Bieniasz |
| Recombinant DNA reagent | A475D | *Schmidt et al., 2020*, and this paper | | pSARS-CoV-2Δ19 containing the indicated mutation. Inquiries should be addressed to P.Bieniasz |
| Recombinant DNA reagent | G476A | *Schmidt et al., 2020*, and this paper | | pSARS-CoV-2Δ19 containing the indicated mutation. Inquiries should be addressed to P.Bieniasz |
| Recombinant DNA reagent | G476S | *Schmidt et al., 2020*, and this paper | | pSARS-CoV-2Δ19 containing the indicated mutation. Inquiries should be addressed to P.Bieniasz |
| Recombinant DNA reagent | T487I | *Schmidt et al., 2020*, and this paper | | pSARS-CoV-2Δ19 containing the indicated mutation. Inquiries should be addressed to P.Bieniasz |
| Recombinant DNA reagent | V483I | *Schmidt et al., 2020*, and this paper | | pSARS-CoV-2Δ19 containing the indicated mutation. Inquiries should be addressed to P.Bieniasz |
| Recombinant DNA reagent | V483A | *Schmidt et al., 2020*, and this paper | | pSARS-CoV-2Δ19 containing the indicated mutation. Inquiries should be addressed to P.Bieniasz |
| Recombinant DNA reagent | V483F | *Schmidt et al., 2020*, and this paper | | pSARS-CoV-2Δ19 containing the indicated mutation. Inquiries should be addressed to P.Bieniasz |
| Recombinant DNA reagent | E484Q | *Schmidt et al., 2020*, and this paper | | pSARS-CoV-2Δ19 containing the indicated mutation. Inquiries should be addressed to P.Bieniasz |
| Recombinant DNA reagent | E484A | *Schmidt et al., 2020*, and this paper | | pSARS-CoV-2Δ19 containing the indicated mutation. Inquiries should be addressed to P.Bieniasz |
| Recombinant DNA reagent | E484D | *Schmidt et al., 2020*, and this paper | | pSARS-CoV-2Δ19 containing the indicated mutation. Inquiries should be addressed to P.Bieniasz |
| Recombinant DNA reagent | F490S | *Schmidt et al., 2020*, and this paper | | pSARS-CoV-2Δ19 containing the indicated mutation. Inquiries should be addressed to P.Bieniasz |

*Continued on next page*

Continued

| Reagent type (species) or resource | Designation | Source or reference | Identifiers | Additional information |
|---|---|---|---|---|
| Recombinant DNA reagent | F490L | *Schmidt et al., 2020*, and this paper | | pSARS-CoV-2Δ19 containing the indicated mutation. Inquiries should be addressed to P.Bieniasz |
| Recombinant DNA reagent | Q493K | *Schmidt et al., 2020*, and this paper | | pSARS-CoV-2Δ19 containing the indicated mutation. Inquiries should be addressed to P.Bieniasz |
| Recombinant DNA reagent | Q493R | *Schmidt et al., 2020*, and this paper | | pSARS-CoV-2Δ19 containing the indicated mutation. Inquiries should be addressed to P.Bieniasz |
| Recombinant DNA reagent | S494P | *Schmidt et al., 2020*, and this paper | | pSARS-CoV-2Δ19 containing the indicated mutation. Inquiries should be addressed to P.Bieniasz |
| Recombinant DNA reagent | N501Y | *Schmidt et al., 2020*, and this paper | | pSARS-CoV-2Δ19 containing the indicated mutation. Inquiries should be addressed to P.Bieniasz |
| Recombinant DNA reagent | V503F | *Schmidt et al., 2020*, and this paper | | pSARS-CoV-2Δ19 containing the indicated mutation. Inquiries should be addressed to P.Bieniasz |
| Sequence-based reagent | endof_M_for | This paper | PCR and sequencing primer | CTATCGGCCACTT CAAATGAGCTAG |
| Sequence-based reagent | L_begin_rev | This paper | PCR and sequencing primer | TCATGGAAGTCCA CGATTTTGAGAC |
| Sequence-based reagent | VSV-RBD-F primer | This paper | PCR and sequencing primer | CTGGCTCTGCACA GGTCCTACCTGACA |
| Sequence-based reagent | VSV-RBD-R primer | This paper | PCR and sequencing primer | CAGAGACATTGT GTAGGCAATGATG |
| Peptide, recombinant protein | ACE2-Fc fusion protein | This paper | | Recombinant ACE2 extracellular domain fused to IgG1 Fc see Materials and Methods Inquiries should be addressed to P.Bieniasz |
| Peptide, recombinant protein | S-6P-NanoLuc | This paper | | A conformationally stabilized (6P) version of the SARS-CoV-2 S protein fused to Nanoluciferase See materials and methods Inquiries should be addressed to P.Bieniasz |
| Commercial assay or kit | Trizol-LS | Thermo Fisher | Cat# 10296028 | |

*Continued on next page*

*Continued*

| Reagent type (species) or resource | Designation | Source or reference | Identifiers | Additional information |
|---|---|---|---|---|
| Commercial assay or kit | Superscript III reverse transcriptase | Thermo Fisher | Cat# 18080093 | |
| Commercial assay or kit | Nextera TDE1 Tagment DNA enzyme | Illumina | Cat# 15027865 | 0.25 µl |
| Commercial assay or kit | TD Tagment DNA buffer | Illumina | Cat# 15027866 | 1.25 µl |
| commercial assay or kit | Nextera XT Index Kit v2 | Illumina | Cat# FC-131–2001 | |
| Commercial assay or kit | KAPA HiFi HotStart ReadyMix | KAPA Biosystems | Cat# KK2601 | |
| Commercial assay or kit | AmPure Beads XP | Agencourt | Cat# A63881 | |
| Commercial assay or kit | Expi293 Expression System Kit | Thermo Fisher Scientific | Cat# A14635 | |
| Commercial assay or kit | Ni-NTA Agarose | Qiagen | Cat# 30210 | |
| Commercial assay or kit | HRV 3C Protease | TaKaRa | Cat# 7360 | |
| Commercial assay or kit | LI-COR Intercept blocking buffer | Licor | P/N 927–70001 | |
| Commercial assay or kit | Dynabeads Protein G | Thermo Fisher Scientific | Cat# 10004D | |
| Commercial assay or kit | Passive Lysis 5X Buffer | Promega | Cat# E1941 | |
| Commercial assay or kit | Nano-Glo Luciferase Assay System | Promega | Cat# N1150 | |
| Software, algorithm | Geneious Prime | https://www.geneious.com/ | RRID:SCR_010519 | Version 2020.1.2 |
| Software, algorithm | Python programming language | https://www.python.org/ | RRID:SCR_008394 | version 3.7 |
| Software, algorithm | pandas | 10.5281/zenodo.3509134 | RRID:SCR_018214 | Version 1.0.5 |
| Software, algorithm | numpy | 10.1038/s41586-020-2649-2 | RRID:SCR_008633 | Version 1.18.5 |
| Software, algorithm | matplotlib | 10.1109/MCSE.2007.55 | RRID:SCR_008624 | Version 3.2.2 |

## Plasmid constructs

A replication-competent rVSV/SARS-CoV-2/GFP chimeric virus clone, encoding the SARS-CoV-2 spike protein lacking the C-terminal 18 codons in place of G, as well as GFP immediately upstream of the L (polymerase) has been previously described (*Schmidt et al., 2020*). The pHIV-1$_{NL}$GagPol and pCCNG/nLuc constructs that were used to generate SARS-CoV-2 pseudotyped particles have been previously described (*Schmidt et al., 2020*). The pSARS-CoV-2 protein expression plasmid containing a C-terminally truncated SARS-CoV-2 S protein (pSARS-CoV-2$_{\Delta 19}$) containing a synthetic human-codon-optimized cDNA (Geneart) has been previously described (*Schmidt et al., 2020*) and was engineered to include BamHI, MfeI, BlpI and AgeI restriction enzyme sites flanking sequences encoding the RBD. Gibson assembly was used to introduce mutant RBD sequences into this plasmid,

that were generated synthetically (g/eBlocks IDT) or by overlap extension PCR with primers that incorporated the relevant nucleotide substitutions.

## Cell lines

HEK-293T cells and derivatives were cultured in Dulbecco's Modified Eagle Medium (DMEM) supplemented with 10% fetal bovine serum (FBS) at 37°C and 5% $CO_2$. All cell lines have been tested negative for contamination with mycoplasma. Derivatives expressing ACE2 were generated by transducing 293T cells with CSIB(ACE2) vector and the uncloned bulk population 293T/ACE2(B) or a single-cell clone 293T/ACE2.cl22 (*Schmidt et al., 2020*) were used.

## Replication-competent VSV/SARS-CoV-2/GFP chimeric virus

The generation of infectious rVSV/SARS-CoV-2/GFP chimeric viruses stocks has been previously described (*Schmidt et al., 2020*). Two plaque-purified variants designated rVSV/SARS-CoV-2/GFP$_{1D7}$ and rVSV/SARS-CoV-2/GFP$_{2E1}$ that encode F157S/R685M (1D7) and D215G/R683G (2E1) substitutions were used in these studies. The rVSV/SARS-CoV-2/GFP chimeric virus was used under enhanced BSL-2 conditions. i.e in a BLS-2 laboratory with BSL-3 like precautions.

## HIV-1/CCNanoLuc2AEGFP-SARS-CoV-2 pseudotype particles

The HIV-1/NanoLuc2AEGFP-SARS-CoV-2 pseudotyped virions were generated as previously described (*Schmidt et al., 2020*). Briefly, 293T cells were transfected with pHIV$_{NL}$GagPol, pCCNanoLuc2AEGFP and a WT or mutant SARS-CoV-2 expression plasmid (pSARS-CoV-2$_{\Delta19}$) using polyethyleneimine. At 48 hr after transfection, the supernatant was harvested, clarified, filtered, aliquoted and stored at −80°C.

## Infectivity assays

To measure the infectivity of pseudotyped or chimeric viral particles, viral stocks were serially diluted and 100 μl of each dilution added to 293T/ACE2cl.22 target cells plated at $1 \times 10^4$ cells/well in 100 μl medium in 96-well plates the previous day. Cells were then cultured for 48 hr (HIV-1 pseudotyped viruses) or 16 hr (replication-competent rVSV/SARS-CoV-2/GFP), unless otherwise indicated, and then photographed or harvested for NanoLuc luciferase or flow cytometry assays.

## Selection of viruses in the presence of antibodies

For selection of viruses resistant to plasma or monoclonal antibodies, rVSV/SARS-CoV-2/GFP$_{1D7}$ and rVSV/SARS-CoV-2/GFP$_{2E1}$ populations containing $10^6$ infectious particles were used. To generated the viral populations for selection experiments,, rVSV/SARS-CoV-2/GFP$_{1D7}$ and rVSV/SARS-CoV-2/GFP$_{2E1}$ were passaged generate diversity, incubated with dilutions of monoclonal antibodies (10 μg/ml, 5 μg/ml) or COVID19 plasma (1:50, 1:250, 1:500) for 1 hr at 37 °C. Then, the virus-antibody mixtures were incubated with $2 \times 10^5$ 293T/ACE2(B) cells in 12-well plates. Two days later, the cells were imaged and supernatant from the wells containing the highest concentration of plasma or monoclonal antibodies that showed evidence of viral replication (GFP-positive foci) or large numbers of GFP-positive cells was harvested. A 100 μl of the cleared supernatant was incubated with the same dilution of plasma or monoclonal antibody and then used to infect $2 \times 10^5$ 293T/ACE2(B) cells in 12-well plates, as before. rVSV/SARS-CoV-2/GFP$_{1D7}$ and rVSV/SARS-CoV-2/GFP$_{2E1}$ were passaged in the presence of C121 or C144 two times before complete escape was apparent. rVSV/SARS-CoV-2/GFP$_{1D7}$ and rVSV/SARS-CoV-2/GFP$_{2E1}$ were passaged with C135 or plasma samples up to five times.

To isolate individual mutant viruses, selected rVSV/SARS-CoV-2/GFP$_{1D7}$ and rVSV/SARS-CoV-2/GFP$_{2E1}$ populations were serially diluted in medium without antibodies and individual viral variants isolated by visualizing single GFP-positive plaques at limiting dilutions in 96-well plates containing $1 \times 10^4$ 293T/ACE2(B) cells. These plaque-purified viruses were expanded, and further characterized using sequencing, spreading replication and neutralization assays.

## Sequence analyses

For the identification of putative antibody resistance mutations, RNA was isolated from aliquots of supernatant containing selected viral populations or individual plaque-purified variants using Trizol-

LS. The purified RNA was subjected to reverse transcription using random hexamer primers and Superscript III reverse transcriptase (Thermo Fisher Scientific, US). The cDNA was amplified using Phusion (NEB, US) and primers flanking RBD encoding sequences. Alternatively, a fragment including the entire S-encoding sequence was amplified using primers targeting VSV-M and VSV-L. The PCR products were gel-purified and sequenced either using Sanger-sequencing or NGS as previously described (*Gaebler et al., 2019*). Briefly, 1 µl of diluted DNA was used for the tagmentation reactions with 0.25 µl Nextera TDE1 Tagment DNA enzyme (catalog no. 15027865), and 1.25 µl TD Tagment DNA buffer (catalog no. 15027866; Illumina). Subsequently, the DNA was ligated to unique i5/i7 barcoded primer combinations using the Illumina Nextera XT Index Kit v2 and KAPA HiFi Hot-Start ReadyMix (2X; KAPA Biosystems) and purified using AmPure Beads XP (Agencourt), after which the samples were pooled into one library and subjected to paired-end sequencing using Illumina MiSeq Nano 300 V2 cycle kits (Illumina) at a concentration of 12pM.

For analysis of NGS data, the raw paired-end reads were pre-processed to remove adapter sequences and trim low-quality reads (Phred quality score < 20) using BBDuk. Filtered reads were mapped to the codon-optimized SARS-CoV-2 S sequence in rVSV/SARS-CoV-2/GFP using Geneious Prime (Version 2020.1.2). Mutations were annotated using Geneious Prime, with a P-value cutoff of $10^{-6}$. Information regarding RBD-specific variant frequencies, their corresponding P-values, and read depth were compiled using the Python programming language (version 3.7) running pandas (1.0.5), numpy (1.18.5), and matplotlib (3.2.2).

## Neutralization assays

To measure neutralizing antibody activity in plasma, serial dilutions of plasma beginning with a 1:12.5 or a 1:100 (for plasma COV-NY) initial dilution were five-fold serially diluted in 96-well plates over six or eight dilutions. For monoclonal antibodies, or an ACE2-IgG1Fc fusion protein the initial dilution started at 40 µg/ml. Thereafter, approximately $5 \times 10^4$ infectious units of rVSV/SARS-CoV-2/GFP or $5 \times 10^3$ infectious units of HIV/CCNG/nLuc/SARS-CoV-2 were mixed with the plasma or mAb at a 1:1 ratio and incubated for 1 hr at 37°C in a 96-well plate. The mixture was then added to 293T/ACE2cl.22 target cells plated at $1 \times 10^4$ cells/well in 100 µl medium in 96-well plates the previous day. Thus, the final starting dilutions were 1:50 or 1:400 (for COV-NY) for plasma and 10 µg/ml for monoclonal antibodies. Cells were then cultured for 16 hr (for rVSV/SARS-CoV-2/GFP) or 48 hr (for HIV/CCNG/nLuc/SARS-CoV-2). Thereafter, cells were harvested for flow cytometry or NanoLuc luciferase assays.

## Antibody-binding and ACE2-binding inhibition assay

A conformationally stabilized (6P) version of the SARS-CoV-2 S protein (*Hsieh et al., 2020*), appended at its C-terminus with a trimerization domain, a GGSGGn spacer sequence, NanoLuc luciferase, Strep-tag, HRV 3C protease cleavage site and 8XHis (S-6P-NanoLuc) was expressed and purified from the supernatant of 293T Expi cells. Mutants thereof were also expressed and purifies following substitution of sequences encoding the RBD that originated from the unmodified S-expression plasmids.

For antibody-binding assays, 20, 40, or 80 ng S-6P-NanoLuc (or mutants thereof) were mixed with 100 ng of antibodies, C121, C135, or C144, diluted in LI-COR Intercept blocking buffer, in a total volume of 60 µl/well in 96-well plate. After a 30 min incubation, 10 µl protein G magnetic beads was added to each well and incubated for 1.5 hr. The beads were then washed three times and incubated with 30 µl lysis buffer (Promega). Then 15 µl of the lysate was used to measure bound NanoLuc activity.

For ACE2-binding inhibition assays, 20 ng of S-6P-NanoLuc was mixed with 100 ng of antibodies, C121, C135, or C144, diluted in 3% goat serum/PBS, in a total volume of 50 µl. After 30 min incubation, the mixture was incubated with $1 \times 10^5$ 293 T cells, or 293T/ACE2cl.22 cells for 2 hr at 4°C. The cells were then washed three times and lysed with 30 µl lysis buffer and 15 µl of the lysate was used to measure bound NanoLuc activity.

## Reporter gene assays

For the NanoLuc luciferase assays, cells were washed gently, twice with PBS and lysed in Lucifersase Cell culture Lysis reagent (Promega). NanoLuc luciferase activity in the lysates was measured using

the Nano-Glo Luciferase Assay System (Promega) and a Modulus II Microplate Multimode reader (Turner BioSystem) or a Glowmax Navigator luminometer (Promega), as described previously (*Schmidt et al., 2020*). To record GFP+ cells, 12-well plates were photographed using an EVOS M7000 automated microscope. For flow cytometry, cells were trypsinized, fixed and enumerated using an Attune NxT flow cytometer. The half maximal inhibitory concentrations for plasma ($NT_{50}$), and monoclonal antibodies ($IC_{50}$) was calculated using 4-parameter nonlinear regression curve fit to raw or normalized infectivity data (GraphPad Prism). Top values were unconstrained, the bottom values were set to zero.

## Human plasma samples and monoclonal antibodies

The human plasma samples COV-47, COV-72 and COV-107 and monoclonal antibodies C144, C135 and C121 used in this study were previously reported (*Robbiani et al., 2020*). The human plasma sample COV-NY was obtained from the New York Blood Center (*Luchsinger et al., 2020*). All plasma samples were obtained under protocols approved by Institutional Review Boards at both institutions.

## Acknowledgements

We acknowledge the generous contributions the GISAID database contributors and curators. This work was supported by NIH grants P01AI138398-S1, 2U19AI111825 (to MCN and CMR), R01AI091707-10S1 (to CMR) R01AI078788 (to TH) R37AI64003 (to PDB) and by the George Mason University Fast Grant (to DFR and to CMR) and the European ATAC consortium EC101003650 (to DFR) and the G Harold and Leila Y Mathers Charitable Foundation (to CMR). DP was supported by a Medical Scientist Training Program grant from the National Institute of General Medical Sciences of the National Institutes of Health under award number T32GM007739 to the Weill Cornell/Rockefeller/Sloan Kettering Tri-Institutional MD-PhD Program. CG was supported by the Robert S Wennett Post-Doctoral Fellowship, in part by the National Center for Advancing Translational Sciences (National Institutes of Health Clinical and Translational Science Award program, grant UL1 TR001866), and by the Shapiro-Silverberg Fund for the Advancement of Translational Research. PDB and MCN are Howard Hughes Medical Institute Investigators.

## Additional information

### Competing interests

Yiska Weisblum: Rockefeller University has applied for a patent relating to the replication compentent VSV/SARS-CoV-2 chimeric virus on which YW is listed as an inventor (US patent 63/036,124). Fabian Schmidt: Rockefeller University has applied for a patent relating to the replication compentent VSV/SARS-CoV-2 chimeric virus on which FS is listed as an inventor (US patent 63/036,124). Davide F Robbiani: Rockefeller University has applied for a patent relating to SARS-CoV-2 monoclonal antibodies on which DFR is listed as an inventor. Michel C Nussenzweig: Rockefeller University has applied for a patent relating to SARS-CoV-2 monoclonal antibodies on which MCN is listed as an inventor. Theodora Hatziioannou: Rockefeller University has applied for a patent relating to the replication compentent VSV/SARS-CoV-2 chimeric virus on which TH is listed as an inventor (US patent 63/036,124). Paul D Bieniasz: Rockefeller University has applied for a patent relating to the replication compentent VSV/SARS-CoV-2 chimeric virus on which PDB is listed as an inventor (US patent 63/036,124). The other authors declare that no competing interests exist.

### Funding

| Funder | Grant reference number | Author |
| --- | --- | --- |
| National Institute of Allergy and Infectious Diseases | R37AI64003 | Paul D Bieniasz |
| National Institute of Allergy and Infectious Diseases | R01AI078788 | Theodora Hatziioannou |
| National Institute of Allergy | P01AI138398-S1 | Charles M Rice |

| | | |
|---|---|---|
| and Infectious Diseases | | Michel C Nussenzweig |
| National Institute of Allergy and Infectious Diseases | R01AI091707-10S1 | Charles M Rice |
| George Mason University | Fast Grant | Davide F Robbiani Charles M Rice |
| European ATAC Consortium | EC101003650 | Davide F Robbiani |
| National Institutes of Health | UL1 TR001866 | Christian Gaebler |
| National Institute of Allergy and Infectious Diseases | 2U19AI111825 | Charles M Rice Michel C Nussenzweig |
| G. Harold and Leila Y. Mathers Charitable Foundation | | Charles M Rice |
| Robert S. Wennett Post-Doctoral Fellowship | | Christian Gaebler |
| Shapiro-Silverberg Fund | Advancement of Translational Research | Christian Gaebler |
| Howard Hughes Medical Institute | Investigators | Michel C Nussenzweig Paul D Bieniasz |
| National Institute of General Medical Sciences | T32GM007739 | Daniel Poston |

The funders had no role in study design, data collection and interpretation, or the decision to submit the work for publication.

## Author contributions

Yiska Weisblum, Fabian Schmidt, Conceptualization, Investigation; Fengwen Zhang, Justin DaSilva, Daniel Poston, Julio CC Lorenzi, Magdalena Rutkowska, Eleftherios Michailidis, Christian Gaebler, Alice Cho, Zijun Wang, Anna Gazumyan, Melissa Cipolla, Marina Caskey, Investigation; Frauke Muecksch, Hans-Heinrich Hoffmann, Marianna Agudelo, Methodology; Larry Luchsinger, Christopher D Hillyer, Resources; Davide F Robbiani, Supervision; Charles M Rice, Michel C Nussenzweig, Theodora Hatziioannou, Conceptualization, Resources, Supervision, Writing - review and editing; Paul D Bieniasz, Conceptualization, Resources, Data curation, Supervision, Funding acquisition, Writing - original draft, Project administration, Writing - review and editing

## Author ORCIDs

Yiska Weisblum https://orcid.org/0000-0002-9249-1745
Fabian Schmidt https://orcid.org/0000-0001-7731-6685
Frauke Muecksch http://orcid.org/0000-0002-0132-5101
Eleftherios Michailidis http://orcid.org/0000-0002-9907-4346
Larry Luchsinger http://orcid.org/0000-0002-0063-1764
Charles M Rice http://orcid.org/0000-0003-3087-8079
Paul D Bieniasz https://orcid.org/0000-0002-2368-3719

## Ethics

Human subjects: Human plasma samples were obtained from volunteers at the New York Blood Center and Rockefeller University Hospital. Informed consent, and consent to publishers obtained. De-identified Plasma samples from the New York Blood Center were obtained under protocols approved by Institutional Review Boards at the New York Blood Center and authorized by donors under informed consent in accordance with federal, state and local laws and regulations which address protection of human subjects in research, including 45 CFR part 46. Plasma samples at the Rockefeller University were collected with Informed consent, and consent to publishers after review by the Rockefeller University IRB protocol number DRO-1006.

## Decision letter and Author response

Decision letter https://doi.org/10.7554/eLife.61312.sa1

Author response https://doi.org/10.7554/eLife.61312.sa2

## Additional files

### Supplementary files
• Transparent reporting form

### Data availability
All data generated or analysed during this study are included in the manuscript and supporting files.

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
