## [Decision Letter]

**Acceptance summary:**

This paper describes the use of replication-competent vesicular stomatitis viruses (VSV) encoding the SARS-CoV-2 spike (S) protein to generate, identify and characterize antibody neutralization-escape mutants. The study provides new insights to SARS-CoV-2 neutralization escape that have important implications for vaccine development and immunotherapy.

**Decision letter after peer review:**

Thank you for submitting your article "Escape from neutralizing antibodies by SARS-CoV-2 spike protein variants" for consideration by *eLife*. Your article has been reviewed by three peer reviewers, and the evaluation has been overseen by a Reviewing Editor and Jos van der Meer as the Senior Editor. The following individual involved in review of your submission has agreed to reveal their identity: David Montefiore (Reviewer #1).

The reviewers have discussed the reviews with one another and the Reviewing Editor has drafted this decision to help you prepare a revised submission.

Summary:

The paper describes the use of replication-competent vesicular stomatitis viruses encoding genes for SARS-CoV-2 spike protein to generate, identify and characterize neutralization-escape variants. The study provides new information on SARS-CoV-2 escape pathways that have important implications for vaccines and immunotherapy. All three reviewers consider the work to have been well performed, the paper to be clearly written and the novel findings appropriate for publication in *eLife*. Two of the reviewers raise the following minor points that should be addressed prior to publication.

Revisions:

1) The ease of selecting neutralization-escape variants is remarkable and suggests a mutation rate that is under-appreciated. Information on the frequency of escape mutations in the original virus stocks would be informative in this regard. The authors mention this was done by NGS (subsection “Isolation and characterization of rVSV/SARS-CoV-2/GFP antibody escape mutants”) but I could not find the data.

2) Why were titrated doses of convalescent plasma but only a single high dose of mAbs used for selection of escape variants?

3) Figure 3 – are the WT curves replotted on multiple panels (3 of the 4)? If so, please mention in the legend.

4) Figure 4B – Were IC_50_s obtained in all cases in 4B? Or do some of the highest values instead represent the highest concentration tested that did not achieve 50% inhibition (i.e. should these be marked as greater then ">")? The text suggests that resistance was defined as >10µg/mL, if such values are on the graph they should be marked or identified in some way.

5) There are caveats to the conclusion that the neutralizing monoclonals only contribute in a "minor way to the overall neutralizing response in that same individual". Alternatively, the selection scheme here could've selected for the simplest or most common escape mutations, but not necessarily the most fit; in particular, polyclonal plasma may favor 1) mutations that retain fitness in the presence of multiple specificities and 2) mutations that have the least impact on viral replication.

These caveats singly or together could be an alternative explanation for the discrepancies between mutations selected by monoclonals versus plasma from the same individual. That is to say, the monoclonal specificities may be biologically relevant in vivo, but escape plays out differently when multiple selection pressures are acting in combination, possibly selecting for variants or combinations of variants that confer resistance more broadly. in vivo will also involve the replicative properties and mutational spectrum of the coronavirus, the involvement of linked compensatory changes or context-specific effects, and the fact that starting populations may be more complex than the homogenous dose of 10e6 particles used here. Similarly, in vitro selection begins at a high dose of the selection mAbs or mix of mAbs, whereas selection in vivo may occur at low or sub-neutralizing concentrations. It would be useful to discuss how selection could play out differently in the in vitro scheme versus the complexities in vivo, and as a possible shortcoming of using a recombinant rhabdovirus (or to put it another way, while this system is very powerful and rapid, the impacts on viral fitness will require future work in the context of a replication competent SARS-CoV-2 isolate). This is not a flaw in the study, but an important item for discussion.

6) Please discuss whether antibodies found in plasma are indicators of antibodies that might be effective in the respiratory tract (is plasma a suitable proxy)?

7) The lack of citations in the Discussion can only be partially explained by the novelty of SARS-CoV-2; there are many general statements for which there is an abundant literature and additional citations should be included – for example, the first 2-3 paragraphs involve general statements about viral evolutionary dynamics and should be supported with citations and/or useful references for the potentially broad readership of this paper. In other places, statements could be supported by reference to other viral systems.

8) Subsection “Isolation and characterization of rVSV/SARS-CoV-2/GFP antibody escape mutants” – "replication [in] the presence of…" and "at last [least] part of".

---

## [Author Response]

Revisions:1) The ease of selecting neutralization-escape variants is remarkable and suggests a mutation rate that is under-appreciated. Information on the frequency of escape mutations in the original virus stocks would be informative in this regard. The authors mention this was done by NGS (subsection “Isolation and characterization of rVSV/SARS-CoV-2/GFP antibody escape mutants”) but I could not find the data.

We apologize for the ambiguous, and apparently misleading, wording in the subsection “Isolation and characterization of rVSV/SARS-CoV-2/GFP antibody escape mutants” (this has been cleaned up in the revised version). The frequencies of escape mutations “determined by illumina sequencing” in this sentence refers to the antibody selected viral populations, not the unselected starting populations. Unfortunately, the error rate of Illumina sequencing is such that it would be extremely difficult to accurately measure the prevalence of low frequency mutations in the unselected VSV/SARS-CoV-2 populations – so we did not attempt it. Given that we selected essentially the same set of mutations in replicate cultures, and that we began with a viral population size of ~1x10^6^ suggests that each of the selected mutations were present at a frequency of >10^-6^. Moreover, given that the error rate of the VSV polymerase is ~10^-5^/base/replication cycle it is not terribly surprising that escape mutations were present in the starting populations of containing 10^6^ members. We have added a short section to the manuscript to clarify our rationale about this point.

2) Why were titrated doses of convalescent plasma but only a single high dose of mAbs used for selection of escape variants?

The monoclonal antibody selections were actually carried out at a high dose, chosen at 10μg/ml (~1000 to 10,000 x IC_50_) so as to neutralize nearly all the antibody susceptible virus in the starting population of ~10^6^ infectious – units. This would enable only a small number of infection events by mAb sensitive virus, and a ‘clean’ selection of the highly resistant virus (plaques were generally seen at the first passage). For plasmas the considerations are different, given the possibility that there may be multiple different neutralizing antibody specificities at multiple different concentrations – escape from an a given antibody present in a plasma might not be observed if other antibodies are present at a sufficient concentration to neutralize that escape variant. Thus a ‘trial and error’ approach, with multiple different dilutions of plasma seemed more appropriate in that case. We have edited the text at appropriate points to indicate the rationale for these experimental designs.

3) Figure 3 – are the WT curves replotted on multiple panels (3 of the 4)? If so, please mention in the legend.

The WT curves that were run concurrently with each mutant set are replotted in some panels. In some cases, the same WT controls served as controls for multiple different mutant sets that were run concurrently. We have edited the legend to clarify this in the revised manuscript.

4) Figure 4B – Were IC_50_s obtained in all cases in 4B? Or do some of the highest values instead represent the highest concentration tested that did not achieve 50% inhibition (i.e. should these be marked as greater then ">")? The text suggests that resistance was defined as >10µg/mL, if such values are on the graph they should be marked or identified in some way.

The values plotted are the highest concentration tested that did not achieve inhibition, rather than IC_50_ values. The reviewer is correct that these should have been designated >10µg/mL rather than 10µg/mL. This oversight has been corrected in the revised manuscript.

5) There are caveats to the conclusion that the neutralizing monoclonals only contribute in a "minor way to the overall neutralizing response in that same individual". Alternatively, the selection scheme here could've selected for the simplest or most common escape mutations, but not necessarily the most fit; in particular, polyclonal plasma may favor 1) mutations that retain fitness in the presence of multiple specificities and 2) mutations that have the least impact on viral replication.These caveats singly or together could be an alternative explanation for the discrepancies between mutations selected by monoclonals versus plasma from the same individual. That is to say, the monoclonal specificities may be biologically relevant in vivo, but escape plays out differently when multiple selection pressures are acting in combination, possibly selecting for variants or combinations of variants that confer resistance more broadly. in vivo will also involve the replicative properties and mutational spectrum of the coronavirus, the involvement of linked compensatory changes or context-specific effects, and the fact that starting populations may be more complex than the homogenous dose of 10e6 particles used here. Similarly, in vitro selection begins at a high dose of the selection mAbs or mix of mAbs, whereas selection in vivo may occur at low or sub-neutralizing concentrations. It would be useful to discuss how selection could play out differently in the in vitro scheme versus the complexities in vivo, and as a possible shortcoming of using a recombinant rhabdovirus (or to put it another way, while this system is very powerful and rapid, the impacts on viral fitness will require future work in the context of a replication competent SARS-CoV-2 isolate). This is not a flaw in the study, but an important item for discussion.

While the points made by the reviewer are correct, we do not think that they falsify our conclusion that “the most potently neutralizing antibodies generated in a given COVID19 convalescent individual may contribute in only a minor way to the overall neutralizing antibody response in that same individual.” Specifically, if a viral mutant is completely resistant to a given monoclonal antibody from a given individual, yet that same viral mutant retains sensitivity to plasma from that same individual, then it is nearly inescapable that other antibodies in the plasma, not the monoclonal antibody, must dominate the neutralizing activity of the plasma. That the plasma selects mutations at a different site in S to that selected by the potent monoclonal antibody from the same individual, is orthogonal supportive evidence that the monoclonal in question does not constitute the major neutralizing activity in the plasma of that individual.

Nevertheless, the points made by the reviewer in terms of the differences in selection pressures that might play out in vivo compared to our in vitro system are certainly worth of discussion and we have done so in the revised manuscript.

6) Please discuss whether antibodies found in plasma are indicators of antibodies that might be effective in the respiratory tract (is plasma a suitable proxy)?

This point is discussed in the revised manuscript. Without actually sampling the antibodies in the respiratory tract, it is difficult to know how similar the antibodies in plasma are to those and respiratory tract. It is known that the immunoglobulin subtypes (e.g. IgA versus IgG) are differentially represented in plasma versus respiratory secretions, and the concentrations of each immunoglobulin subtype or specificity precisely at the site of replication is unknown. However, it is known that IgG that dominates plasma immunoglobulins is also present in respiratory secretions albeit at lower levels than IgA. Moreover, our recent work has indicated that at least some of the neutralizing activity in plasma in contributed by IgA and several of the antibody lineages that we have cloned from SARS-CoV-2 convalescents are class switched to both IgA and IgG. Overall it seems likely that the antibody specificities present in the respiratory tract are likely to broadly reflect, but perhaps not precisely recapitulate, those present in plasma.

7) The lack of citations in the Discussion can only be partially explained by the novelty of SARS-CoV-2; there are many general statements for which there is an abundant literature and additional citations should be included – for example, the first 2-3 paragraphs involve general statements about viral evolutionary dynamics and should be supported with citations and/or useful references for the potentially broad readership of this paper. In other places, statements could be supported by reference to other viral systems.

We have gone through the Discussion and included additional citations wherever we thought appropriate in the revised version.

8) Subsection “Isolation and characterization of rVSV/SARS-CoV-2/GFP antibody escape mutants” – "replication [in] the presence of…" and "at last [least] part of".

Corrected in the revised version.